# The genesis of Hurricane Nate and its interaction with a nearby environment of very dry air

Blake Rutherford [1], Timothy Dunkerton [1], Michael Montgomery [2], and Scott Braun [3]

[1]Northwest Research Associates, Redmond, WA
[2]Naval Postgraduate School, Monterey, California
[3]NASA

*Correspondence to:* Blake Rutherford (blake@nwra.com)

**Abstract.** The interaction of a tropical disturbance with its environment is thought to play an important role in whether a disturbance will develop or not. Most developing disturbances are somewhat protected from the intrusion of environmental dry air at mid-levels. For African easterly wave (AEW) disturbances, the protective boundary is approximated by closed streamlines in the wave-relative frame, and their interior is called the wave-pouch. The dynamic and thermodynamic processes of spin-up occur inside the pouch.

In this study we define the kinematic boundaries for a non-AEW disturbance in the Bay of Campeche that originated along a sharp frontal boundary in a confluent region of low pressure. We examine these boundaries during the genesis of Hurricane Nate (2011) to show how a pouch boundary on isobaric levels in the Lagrangian frame may allow for some transport into the pouch along the frontal boundary while still protecting the innermost development region. This result illustrates a generic property of weakly unsteady flows, including the time-dependent critical-layer of AEWs, that lateral exchange of air occurs along a segment of the boundary formed by the instantaneous, closed translating streamlines.

Transport in the Lagrangian frame is simplest when measured with respect to the stable and unstable manifolds of a hyperbolic trajectory, which are topologically invariant. In this framework, an exact analysis of vorticity transport identifies the primary source as the advection of vorticity through the entrainment and expulsion of bounded material regions called lobes. We also show how these Lagrangian boundaries impact the concentration of moisture, influence convection, and contribute to the pouch vertical structure.

## 1  Introduction

### a. The cyclogenesis problem

The question of development versus non-development of tropical disturbances is a complex problem that has seen significant interest yet has an inherently high amount of unpredictability. There are many known factors that influence development, such as sea-surface temperatures, available moisture and vorticity, vertical wind shear, and the timing and distribution of convection, see e.g. Gray (1968) and McBride and Zehr (1981). Additionally, synoptic flow features often facilitate development by creating a favourable kinematic and thermodynamic environment where amplification of cyclonic vorticity can occur.

As seen by Frank (1970), most Atlantic tropical cyclones form along an African easterly wave (AEW). Along the wave trough, cyclonic vorticity is amplified by intense convection. The marsupial paradigm predicts the location for genesis to occur at the intersection of the wave-trough axis and wave critical layer where mean flow and wave phase speeds are equal (Dunkerton et al. (2009), hereafter DMW09). In the wave-relative frame of reference, a region of closed circulation, called the pouch, protects the embryonic vortex from adverse environmental conditions and the lateral intrusion of dry air and vertical wind shear. Inside the pouch, air is repeatedly moistened by convection while the parent wave is enhanced by diabatically amplified meso-scale eddies within the wave.

It is at the pouch boundary that the interaction of the proto-vortex with its environment occurs, and transport of any air across the boundary alters the physics within the pouch. When mixed into a vortex, dry air may quench convection reducing the total latent heat release and subsequent convergence, thus reducing the rate of or preventing spin-up of the vortex, see Kilroy and Smith (2013). The role of this dry air and to what extent it is important are still relatively poorly understood due to the fact that entrainment and the definitions of the boundaries themselves are not well defined. In this study, we attempt to define the boundaries more rigorously so that the physical interactions between the pouch and environment can be better studied.

Permeability of the pouch boundary allows environmental air to enter a disturbance, which may prevent development if enough dry air reaches the circulation center, as was shown for Gaston (2010) by Rutherford and Montgomery (2012) and Freismuth (2016), a named tropical storm which was inhibited by dry air and vertical wind shear. Davis and Ahijevych (2012) and Montgomery et al. (2012) found that the pouch became increasingly shallower as the dry air intruded. However, Braun et al. (2012) found that if dry air is only partially entrained and does not reach the center, development may still occur, though the rate of intensification may be reduced.

A pouch whose boundary is open to transport on one side may also favor development, as Lussier et al. (2015) showed in the genesis of Hurricane Sandy (2012), if relatively moist environmental air is entrained into the pouch. In that paper it was shown that the pre-Sandy disturbance in the Caribbean was contiguous to the South American Convergence Zone (SACZ) on its equatorward side, and a direct kinematic pathway existed prior to storm formation, tapping the mid-level moisture of the SACZ. Fortunately for development, the pouch boundary was well-defined and closed on the northern side, sheltering from dry exterior air.

In AEW flows, the pouch boundary enclosing a region of recirculation can be seen by assuming a steady flow in the co-moving frame of the parent wave with $\hat{\mathbf{u}} = \mathbf{u} - \mathbf{c}$ as in DMW09[1]. In this frame, hyperbolic stagnation points $\mathbf{x}_{sp}$ satisfying $\hat{\mathbf{u}}(\mathbf{x_{sp}}) = 0$ appear along the wave's critical layer. The streamlines emanating from the stagnation points in the direction of the eigenvector of the negative/positive eigenvalue of the velocity Jacobian are called the stable and unstable manifolds, see e.g. Ottino (1990), and delineate the inner recirculating flow and open flow of the environment. In contrast to disturbances where the AEW provides the distinguished reference frame and antecedent kinematic structure, disturbances not originating from an AEW or other monochromatic tropical wave pathway do not have a distinguished frame of reference in which kinematic boundaries can be properly defined. These situations can be more complex, involving a combination of different disturbance

---

[1]The vector $\mathbf{c}$ refers to the horizontal translation velocity of a (presumably synoptic-scale) disturbance embedded in a "mean flow" suitably defined, denoted by $\overline{\mathbf{u}}$, a horizontal flow that may vary in space, i.e., have shear in any dimension. The c-vector also may vary in any dimension, but it rarely matches $\overline{\mathbf{u}}$.

types, possibly moving in different directions. In Nate (2011), a quasi-stationary tip rollup occurred at the SW end of a vorticity strip stretching across the Gulf of Mexico to connect with spiral bands emanating from the southwest of Tropical Storm Lee (2011), a storm moving eastward at the time. AEW flows are also more closely approximated by a 2D representation on a constant pressure level, while non-AEW flows may have a more significant vertical component in cases where baroclinicity is important.

In this study, we consider the kinematic boundaries and their impermeability with respect to advection on constant pressure levels and three-dimensional non-advective fluxes for non-AEW disturbances that form along a frontal boundary. We show that the boundary limits the advection of environmental dry air to that contained within a single closed material region plus non-advective fluxes, those fluxes not proportional to horizontal velocities, along the boundary. We extend the wave critical layer theory and its associated translating critical points and manifolds for two reasons. First, there is no distinguished frame of reference provided by the AEW, and second, the time-dependence of the flow causes trajectory paths to cross Eulerian streamlines in any translating or rotating frame.

**b. Hurricane Nate (2011)**

A surface low formed along a frontal boundary in the Bay of Campeche on Sept. 6 and the National Hurricane Center classified this disturbance as a tropical storm on Sept. 7. The pre-Nate pouch was first identified by the Montgomery Research Group on Sept. 6 as P25L[2]. One or more vorticity filaments extending across the Gulf of Mexico from the predecessor, Tropical Storm Lee, connected to Nate's region of formation. These filaments were associated with a strong horizontal gradient of water vapor orthogonal to the frontal boundary. An obvious question arises as to whether the "anti-fuel" behind the frontal zone would affect Nate adversely, or if no adverse effects ultimately were seen, why not? During the development of Nate, the frontal zone itself was deformed into a graceful S-curve by the combined action of Lee and Nate at opposite ends of the frontal zone.

Over the next few days, dry air to the north of the frontal boundary, with ECMWF relative humidity values less than $20\%$ throughout the mid-troposphere, was in close proximity to Nate, yet Nate was still able to intensify. Since satellite visible imagery indicated that dry air remained approximately 1 degree from the storm center, we question whether, and to what extent, environmental dry air reached the core. Factors helping the development of Nate included sea surface temperatures (SST)[3] greater than 29 C and little vertical wind shear. After strengthening briefly to hurricane status for 6-12 hours on Sept. 8[4], Nate weakened to a tropical storm on Sept. 9 after showing mid-level dry air and cooler ocean temperatures (SST$<$27 C) presumably from up-welling created by the stationary storm. After Nate began to track westward, it briefly began to intensify on Sept. 10 before making landfall as a tropical storm in central Mexico on Sept. 11. A summary of Nate is given in Avila and Stewart (2013).

In this study, we examine the 2D Lagrangian flow structures on isobaric surfaces from ECMWF model analyses to describe the transport of dry air and evolution of vorticity at mid-levels. The Lagrangian manifolds defined in the upcoming section

---

[2]The pre-Nate pouch was called P25L as the 25th pouch of the 2011 Atlantic season. This name is used to show the location of the storm for the remainder of this paper.

[3]Sea surface temperature data is from the NCDC Optimum Interpolation Sea Surface Temperature (OISST) .25 degree resolution data set

[4]http://weather.unisys.com/hurricane/atlantic/2011/NATE/track.dat

indicate what flow features, including the remnants of Tropical Storm Lee, contributed to the circulation of Nate and measure the impact of dry air that Nate interacted with after genesis. The Lagrangian boundaries are also shown in relation to regions of convection to show that convection is typically located interior to Lagrangian boundaries.

### c. Outline

The outline of the remainder of this paper is as follows. Section 2 provides an introduction to the mathematical methods for the location of Lagrangian boundaries. In Section 3, we show numerical details of the computations and data sets. In Section 4, we describe the genesis of Nate from the perspective of the evolution of Lagrangian manifolds. In Section 5 we provide a detailed description of the interaction of Nate with its environment, and show how the Lagrangian flow boundaries offer both protection from the outer environment, and help to concentrate vorticity into the vortex core. Conclusions and a discussion of
future work are provided in Section 6.

## 2    A review of stable and unstable manifolds of a hyperbolic trajectory and lobe transport

In generic time-dependent flows in a distinguished frame of reference, flow boundaries are a set of distinguished material curves called the stable and unstable manifolds of a hyperbolic trajectory, see e.g. Ottino (1990) and Samelson and Wiggins (2006). Trajectories of the flow satisfy

$$\dot{\mathbf{x}} = \mathbf{u}(\mathbf{x}, t), \tag{1}$$

where $\mathbf{u} = (u, v)^T$ is the fluid velocity an At instantaneous snapshots, the stability of an air parcel is determined by the eigenvalues of its linearizedd $\mathbf{x} = (x, y)^T$ is the Earth-relative particle location. velocity field. The Okubo-Weiss parameter (OW) accounts for shear and strain to distinguish solid-body rotation from parallel shear flow or stretching regions. Imaginary eigenvalues (or complex conjugate pairs when horizontal divergence is non-zero) of $\nabla \mathbf{u}$, indicate elliptic stability or rotation-
dominated flow, and occur when $OW = \frac{1}{4}(\zeta^2 - S_1^2 - S_2^2) > 0$ where relative vertical vorticity is $\zeta = v_x - u_y$, and $S_1 = u_x - v_y$ and $S_2 = v_x + u_y$ are the strain rates. We have not included divergence in the computation of OW since is approximately an order of magnitude smaller than vorticity in the data used for this study. Hyperbolicity is marked by real eigenvalues of opposite signs of the velocity Jacobian, i.e. $OW < 0$. A hyperbolic trajectory $\mathbf{x}_h(t)$ is a trajectory that remains hyperbolic for all $t$. The stable and unstable manifolds manifolds of $\mathbf{x}_h(t)$ are its attracting sets forward and backward in time,

$$S(\mathbf{x}_h(t)) = \{\mathbf{x} : \mathbf{x}(t) \to \mathbf{x}_h(t), t \to \infty\} \tag{2}$$
$$U(\mathbf{x}_h(t)) = \{\mathbf{x} : \mathbf{x}(t) \to \mathbf{x}_h(t), t \to -\infty\} \tag{3}$$

A hyperbolic trajectory has local subspaces, $S_{loc}(\mathbf{x}_h(t))$ and $U_{loc}(\mathbf{x}_h(t))$ that are tangent to the eigenvectors of the negative/positive eigenvalue of the Jacobian. Therefore, the stable and unstable manifolds are the material curves which are initially $U_{loc}(\mathbf{x}_h(t))$ and $S_{loc}(\mathbf{x}_h(t))$, and can be computed by the advection of the local initial segments. The hyperbolic trajectory
may be located near an Eulerian saddle chosen in a proper reference frame, leading to numerical algorithms for their location,

see Ide et al. (2002) and Mancho et al. (2003). In forward time, particles along hyperbolae adjacent to the stable/unstable manifolds are repelled/attracted to these manifolds. The attracting property of the unstable manifold leads to intense gradients of active and passive scalar tracers, a prominent feature in the atmospheric case examined here.

Due to the difference in the direction of time in the stable and unstable cases, the manifolds may cross at points other than hyperbolic stagnation points, forming material regions, called lobes, that are enclosed by multiple manifold segments. The time-evolution of lobes describes the transport of material "across" the Eulerian boundary and is called lobe dynamics, see Wiggins (2005), Duan and Wiggins (1996), Malhotra and Wiggins (1998). The equivalent term "lobe transport" is introduced by Wiggins (2005) and highlights the role of lobes in altering the distribution of active and passive scalars in geophysical flows. Hyperbolic trajectories, their stable and unstable manifolds, and lobe transport have been applied to many geophysical flows, including Koh and Plumb (2000), Joseph and Legras (2002), Rogerson et al. (1999), Miller et al. (1997), Branicki et al. (2011), Mancho et al. (2006b), Malhotra and Wiggins (1998), Wiggins and Ottino (2004), Duan and Wiggins (1996), Koh and Legras (2002), Wiggins (1992), Rom-Kedar et al. (1990). Lobe transport has also been applied by Rodrigue and Eschenazi (2010) to flows with similar flow boundaries as those in pre-genesis cases, including a Kelvin-Stuart cat's eye flow. In tropical cyclones, lobe dynamics allows one to quantify the net entrainment of relatively dry (and hence low entropy) air, the so-called "anti-fuel" of the hurricane problem.

Boundaries of physically important regions in time-dependent flows may be formed by connected stable and unstable manifold segments that form an enclosure called a separatrix[5]. As the flow evolves with time, the separatrix is redefined as a different set of manifold segments at a later time so that it remains most similar to the expected physical boundary, see e.g. Rom-Kedar et al. (1990).

In the case of Rossby-wave critical-layer flows, a cat's eye region of recirculation is expected (Benney and Bergeron, 1969), which governs not only the kinematics but also the dynamical redistribution of vorticity within the cat's eye and simultaneous reflection or over-reflection of incident Rossby waves (Killworth and McIntyre, 1985). In such cases, the "expected physical boundary" corresponds to the separatrix surrounding the cat's eye. A suitable generalization for unstable Rossby waves on a vortex strip is to imagine that the separatrix becomes wider with time, prior to the emergence of distinct gyres and possible vortex pairing (Rutherford and Dunkerton, 2017). While it is beyond our immediate scope to identify an expected physical boundary in the formation of Nate (2011) along its antecedent vortex strip, our analysis of manifolds suggests unequivocally that such a physical boundary exists.

---

[5]In idealized theoretical models of nonlinear critical-layer flows, the cat's eye boundary is a separatrix of total (wave + mean) stream function inside of which absolute vorticity is advected passively to leading order, see e.g. Killworth and McIntyre (1985); Samelson and Wiggins (2006), and references therein. Elsewhere we refer to this stream function geometry as a "wire-frame" induced by the superposition of wave and mean shear: that is, the result of wave propagation prior to nonlinear overturning of vorticity and passive scalars in the cat's eye (see Figure 1 of DMW09). Lagrangian manifold growth tends to parallel the existing wire-frame, as absolute vorticity is a dynamical tracer. This description is appreciated best in slowly varying flows. By the same token, lobe dynamics is most significant when fast but small background oscillations are superposed on the slower wave, mean-flow interaction. In rapid or highly transient developments, a crisp distinction of interior and exterior flows may not have time to materialize, in which case the vortex core is largely unprotected prior to shear sheath formation. In the absence of hostile influences, storm formation is possible without a separatrix, given enough time, but such simulated developments are unrealistic, as are the underlying assumptions of these spontaneous aggregation experiments.

A schematic of the time-evolution of manifolds of hyperbolic trajectories $H_1$ and $H_2$ in a cat's eye is shown in Figure 1 at times $t_1$ (a) and $t_2$ (b). The stable manifolds $S_1$ and $S_2$ are shown in magenta and red while the unstable manifolds $U_1$ and $U_2$ are shown in blue and cyan. The intersection points of stable and unstable manifolds at points other than $H_1$ and $H_2$ are labeled $I_k$ for $k = 1$ to $k = 6$. The separatrix for the time-dependent cat's eye flow is the combination of stable and unstable manifold segments that most represent an Eulerian cat's eye, and is defined at $t_1$ as $V(t_1) = U_1(H_1, I_1) \cup S_2(I_1, H_2) \cup U_2(H_2, I_6) \cup S_1(I_6, H_1)$. By time $t_2$, the separatrix is defined as $V(t_2) = U_1(H_1, I_3) \cup S_2(I_3, H_2) \cup U_2(H_2, I_4) \cup S_1(I_4, H_1)$ to maintain a cat's eye shape. The lobes labeled $L_1$ and $L_3$ are transported from outside the separatrix to inside the separatrix and lobes $L_2$ and $L_4$ are transported from inside the separatrix to outside the separatrix.

The change in system circulation due to the advection of each lobe may be computed using Stokes' theorem along the lobe boundary. Haynes and McIntyre (1987) and Haynes and McIntyre (1990) find that as a consequence of the impermeability principle, isobaric vertical vorticity is conserved on isobaric surfaces, implying that the circulation within the pouch and lobes is conserved with respect to advective fluxes, and all changes to the pouch circulation from advection are due to lobe transport across the expected physical boundary surrounding the "pouch", so defined. Non-advective fluxes, or those not proportional to horizontal velocities resulting from tilting, friction, and sub grid-scale forces, may act across any material curve including the pouch boundary and across the lobe boundaries, see Haynes and McIntyre (1987).

Since the unstable manifold is attracting under a forward time integration, tracer-like quantities such as equivalent potential temperature ($\theta_e$) and ozone tend to develop strong horizontal gradients along the unstable manifold. If moist convection is located interior to the unstable manifold in the region of confluence of $\theta_e$, i.e. on the side of the manifold with higher $\theta_e$, then these boundaries influence not only the two-dimensional vorticity aggregation, but also vorticity amplification through isobaric convergence caused by three-dimensional stretching in moist convection.

In their mature stage, tropical storms exhibit an additional inner pouch boundary as a Rankine-like vortex core with solid-body rotation is isolated from the pouch exterior by a ring of strong differential rotation (azimuthal shearing), see Rutherford et al. (2015). It is a quasi-circular region of intensely negative Lagrangian OW parameter in stark contrast to the positive OW vortex core, (Rutherford et al. (2015)). The "shear sheath", as we will call it, is effectively a boundary to particle transport and protects the vortex core from interaction with the environment, allowing the vortex to become self-sustaining. These shear sheaths and vortex cores have been observed in many geophysical flows, e.g. McWilliams (1984), Beron-Vera et al. (2010).

While the Lagrangian manifolds characterize the pouch as semi-permeable to transport, whether these intrusions disrupt the internal processes is determined by the existence of an additional boundary between the core and the cat's eye boundary called a shear sheath. In most developing disturbances, the region of high vorticity in the center behaves as a finite-time KAM torus, see Kolmogorov (1954); Haller (2015) with little straining deformation, no mixing, and in solid-body rotation. Particles in the core act coherently as a vortex, meaning that they have the same Lagrangian averaged rotation rate, see Haller (2015). A Lagrangian vortex is a convex set with positive values of the Lagrangian averaged vorticity field, defined as

$$\zeta_{Lag}(t_0, t) = \int_{\mathcal{I}} \zeta(\mathbf{x}(s)) ds \tag{4}$$

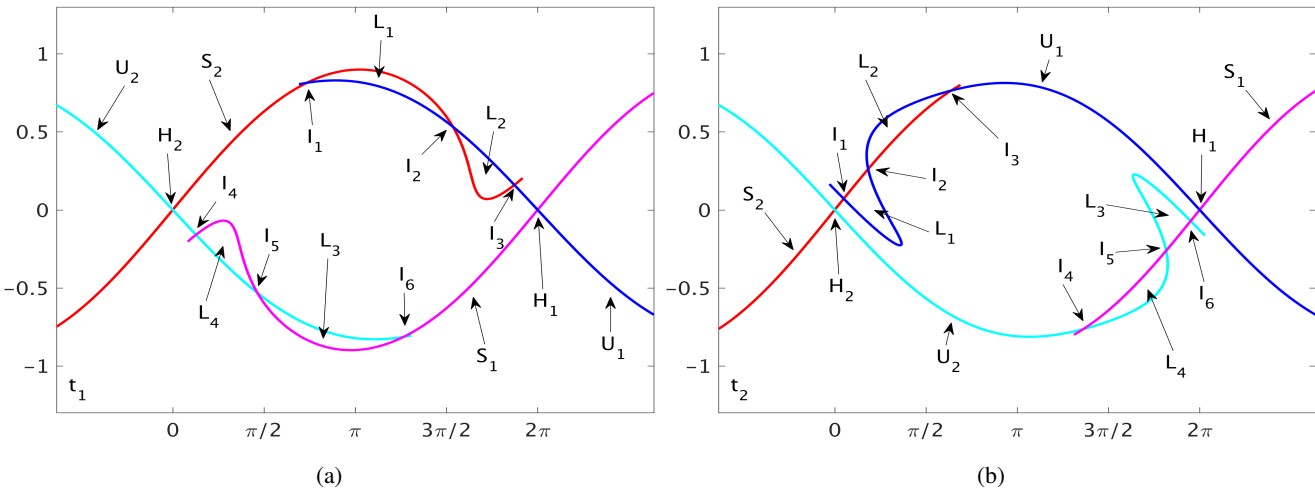

**Figure 1.** Diagram of lobe transport for a time-dependent cat's eye flow where the contents of the lobes are transported across the separatrix boundary formed by the unstable (blue and cyan) and stable (red and magenta) manifolds of a pair of hyperbolic trajectories. The lobes labeled '$L_1$' and '$L_3$' begin outside the cat's eye and are transported to inside the cat's eye.

where $\mathbf{x}(t)$ is the particle trajectory initialized at the point $(\mathbf{x_0}, t_0)$ and trajectories are initialized as a grid and then advected using the two-dimensional velocity field over the time interval $\mathcal{I} = (t_0 - t_1, t_0 + t_2)$. The vortex core can be recognized by nearly circular contours with the highest values of $\zeta_{Lag}$ near the center of the vortex. The lack of mixing in the core is a well known fluid dynamical concept, having been observed in many flows even when there is permeability of the outer boundary, see

5   e.g. Babiano et al. (1994); Lapeyre (2002). Outside of the KAM-torus or vortex core, a region of exceptionally high straining called the shear sheath provides protection to the core as a dynamical barrier with high vorticity gradient, and creates filaments chaotically of parcels that attempt to enter into it. Following Rutherford et al. (2015), the shear sheath can be seen as a minimal annulus of the Lagrangian OW field defined by

$$OW_{Lag} = imag(G) - real(G), \tag{5}$$

10   where $G$ is formed by time-integrating the eigenvalue $\lambda_+$ of the velocity gradient tensor $\nabla \mathbf{u}$ along particle trajectories $\mathbf{x}(t)$,

$$G = \int_{\mathcal{I}} \lambda_+(\mathbf{x}(t), t)dt. \tag{6}$$

$\mathcal{I}$ is the time interval of interest, here a sliding 72 hour time interval, $\mathcal{I} = (t - 36, t + 36)$. Just as lobe dynamics was seen during genesis, a vortex core and shear sheath surrounding it are present just after genesis, and these structures define the Lagrangian topology once there has been significant folding near the hyperbolic trajectories.

### 3    Numerical methods and data

#### a. ECMWF model output

For this study, we use operational ECMWF model analyses constructed at the start of each assimilation cycle to initialize forecast models with velocities and thermodynamic variables given on a regular 0.25×0.25 degree grid every 6 hours.

To compare the ability of the ECMWF analyses to correctly represent the Lagrangian topology with 6 hour data, we also use WRF simulations at 10 km horizontal grid-spacing with 10 minute output intervals. The WRF simulations were initialized with 25 km ECMWF analyses. The ECMWF analyses were used to control the boundary conditions for the entire simulation.

#### b. Computation of fluid velocities and trajectories

Trajectory computations use isobaric velocities with bi-cubic interpolation in space and time. Manifold computations and Lagrangian scalar fields come from sets of particle trajectories, which are computed using a fourth order Runge-Kutta method with an intermediate time step of 15 minutes for the ECMWF trajectories and at the model output timestep of 10 minutes for the WRF simulations to accurately represent the curvature of particle paths.

#### c. Manifold computations

Manifolds in time-dependent flows require the location of a hyperbolic trajectory and its local stable and unstable manifold segments. The initial segment used to construct Lagrangian manifolds is typically a line segment that straddles a hyperbolic trajectory. Hyperbolic trajectories are difficult to locate prior to manifold computation since they require first identifying a distinguished frame of reference and then locating quasi-steady saddle points in that frame. Alternatively, material surfaces called Lagrangian coherent structures (LCSs) may still be found which strongly influence nearby trajectories. If LCSs can be located, manifold segments are initialized along attracting LCSs found in a forward direction in time for the unstable manifold segments and in a backward direction in time for the stable manifold segments. Ridges of the finite-time Lyapunov exponent (FTLE) field have been used for initial segment location for the polar vortex by Koh and Legras (2002). Such initial segments can, in fact, be relatively long compared to a typical initial segment, having acquired already the distended shape of an unstable manifold prior to further advection and distention after initialization. Here, we use the strainline approach of Farazmand and Haller (2012) for initial segment location, where local attracting manifold segments are found to be lines that contain a maxima of the greater eigenvalue of the Cauchy-Green deformation gradient tensor, and are everywhere tangent to the eigenvector field associated with the smaller eigenvalue. The Cauchy-Green deformation tensor is defined as $C = (\nabla F)^T (\nabla F)$ where $F = d\mathbf{x}/d\mathbf{x}_0$ is the Lagrangian deformation tensor and $T$ is the matrix transpose. The tensors $C$ and $F$ have an integration time associated with them, which is relatively short compared to the entire time interval under which genesis occurs, chosen here as 24 hours forward from the initial time and 24 hours backward from the final time. Locating initial segments may also be done using Eulerian strainlines, as defined by Serra and Haller (2016). A choice of 24 hours was long enough to eliminate

spurious Eulerian strainlines while limiting excessive filamentation of the initial segment. Once the manifolds are known, hyperbolic trajectories can be deduced as the intersection of the stable and unstable manifolds.

Manifolds are advected using the algorithms of Mancho et al. (2003) and Mancho et al. (2006a). As the manifolds are advected, the entire set of points evolves, and additional points are inserted when adjacent points grow too far apart. Lagrangian manifolds are advected for the finite time interval during which the hyperbolic trajectories are known to exist. The value of $t$ is chosen at the *beginning* of a sliding time window for the *unstable* manifold (obtained from forward trajectories) and the *end* of the interval for the *stable* manifold (obtained from backward trajectories). This time interval is designed to optimize the description of Lagrangian manifolds of finite duration, and consists of fixed, discrete, non-overlapping windows (to highlight stages of development, as in this paper) or as overlapping windows that slide forward automatically in time (as in the Montgomery Research Group pouch products, when the stages of development are not yet identified in near real-time). The choice of start time and end time for manifold integration change the manifold length, as longer integration times produce longer manifolds and more lobes. Manifolds integrated for shorter times may not produce a closed separatrix or lobes. Our choice of integration time is driven by the objective of capturing the entrainment of the dry air while minimizing the number of additional lobes.

## 4    Genesis of Nate

Tropical Storm Lee left the Gulf of Mexico on 3 Sept., making landfall in Louisiana and traveling northeast. From 4 Sept. to 6 Sept., southerly flow in the Gulf of Mexico guided moisture and remnant vorticity from Lee into the Bay of Campeche where a frontal boundary between moist air to the south and very dry air to the northeast was established. The potential vorticity field showing the pre-Nate region and TS Lee is shown at 0 and 1200 UTC 5 Sept. in Figure 2 and at 0 UTC 6 Sept. in Figure 3 (a). This moisture and vorticity accumulated in a confluent region of low pressure in the southern Gulf of Mexico. Over the next day, the area of low pressure became better organized and showed curved banding features, increased convection, and a well-defined low-level circulation, prompting NHC to initiate the system as Tropical Storm Nate at 2100 UTC 7 Sept., with a conservative 40 kt wind speed estimate taken from oil rig and aircraft measurements[6].

In the nearly stationary frame of the 700 hPa Nate flow in ECMWF model analyses, there was no pouch at any level on 5 Sept. However, there is a saddle point visible on 4 Sept. at 500 hPa in the frame of reference moving with Tropical Storm Lee to the north at $c_y = 5$ m/s (not shown). The saddle point in the nearly stationary frame of Nate emerges at progressively lower levels, emerging at 700 hPa at 1200 UTC 6 Sept., and its streamlines form an enclosed Eulerian pouch at 0 UTC 7 Sept.

### a. Manifolds and lobe transport

We consider now the period of formation by analyzing the Lagrangian manifolds from 0 UTC Sept. 6 to 0 UTC Sept. 9 to see the sources of vorticity and the establishment of a pouch boundary as a barrier to very dry air. The manifolds are overlaid on the $\theta_e$ field at 700 hPa for this time period in Figure 3 (b)-(g). The manifolds and pouch region are labeled in (b), while the

---

[6]National Hurricane Center Tropical Storm Nate Discussion Number 1. http://www.nhc.noaa.gov/archive/2011/al15/al152011.discus.001.shtml

time evolution is shown at 12 hour snapshots in (b)-(g). The stable manifolds $S_1$ and $S_2$ are shown as magenta and red curves, respectively, while the unstable manifolds $U_1$ and $U_2$ are shown as blue and cyan curves, respectively. The initial segments of $U_1$ and $U_2$ were located as strainlines at 0 UTC Sept. 6 and were advected forward in time, while the initial segments of $S_1$ and $S_2$ were strainline segments from 0 UTC Sept. 9 that were advected backward in time. From the intersections of these manifolds, we can deduce the locations of two hyperbolic trajectories, labeled $H_1$ and $H_2$ in Figure 3 ( b). $H_1$ is the intersection of $U_1$ (blue) and $S_1$ (magenta) at the northeast of the pouch, and $H_2$ is the intersection of $U_2$ (cyan) and $S_2$ (red) at the southwest of the pouch. These hyperbolic trajectories are in close proximity to a pair of Eulerian saddles (not shown) that emerge along the regions of confluence on 6 Sept. and are trackable until 10 Sept. There are additional intersection points between the manifolds as $I_1$ and $I_2$ mark the intersections of $U_1$ and $S_2$. Similarly, $U_2$ and $S_1$ intersect at $I_4$. As defined in the mathematical methods section, we can define the pouch as the interior of the connected curves $U_1(H_1, I_1) \cup S_2(I_1, H_2) \cup U_2(H_2, I_4) \cup S_1(I_4, H_1)$.

The unstable manifolds comprising the pouch boundary are attracting regions that can actually be observed before Sept. 6. On 5 Sept., there are two attracting lines in the confluent region from the Lee flow, one from the southern side that is already elongated at 0 UTC and one from the northern side that emerges as a short segment at 1200 UTC. These lines are shown as blue and yellow curves, respectively, in Figure 2 . The origin of $U_1$ is the attracting line from the northern side that is advected southward and becomes the obvious dry air boundary by Sept. 6. Additional attracting lines are advected southward from the Lee flow. Most importantly, one of these lines differentiates the region of the pouch interior that was advected from Lee versus the part that came from the Nate development region. The location of this line can be seen as the yellow curve in Figure 3 (a) that divides the pouch region into $R_1$ to the north of the yellow curve with origins from Lee and $R_2$ the portion of the pouch with southern Gulf of Mexico origins. The advection of this line can be seen at 600 UTC 7 Sept. in Figure 3 (d), as this curve begins to become wrapped in the core as $R_1$ and $R_2$ become mixed. The different properties of these regions can also be seen in the latitude tracer field in Figure 4 (a), which shows the initial latitude of trajectories advected backward 48 hours from the gridded locations at 1800 UTC Sept. 6. The contents of the pouch at this time coming from Lee $(R_1)$ have initial latitudes of greater than 30 degrees, while the portion with Gulf of Mexico origins $(R_2)$ have latitudes less than 30 degrees. We see that the pouch region that originated from Lee still contains high potential vorticity air, Figure 4 (b). The difference between these regions is also clearly visible in the ozone mixing ratio, Figure 4 (e), where higher values are present in $R_1$. The third attracting curve is located over Mexico and the southern Gulf of Mexico, and becomes $U_2$, the southern boundary of Nate during the genesis period. The stable manifold segments $S_1$ and $S_2$ can be located on 0 UTC Sept. 9.

On 1200 UTC Sept. 5, the area of $R_1$ is $1.5 \times 10^5 km^2$, while the area of $R_2$ is $2.8 \times 10^5 km^2$, with a total area of $4.3 \times 10^5 km^2$. As $R_1$ and $R_2$ are mixed, convergence causes a decrease in their areas to $1.3 \times 10^5 km^2$ and $2.3 \times 10^5 km^2$ for a combined area of $3.6 \times 10^5 km^2$ by 1200 UTC 7 Sept. The circulation of $R_1$ is $3.4 \times 10^6 m^2/s$ at 1200 UTC 5 Sept. and increases to $5.3 \times 10^6 m^2/s$ by 1800 UTC 5 Sept. before steadily decreasing to $3 \times 10^6 m^2/s$ by 1200 UTC 7 Sept.[7] The circulation of $R_2$ steadily increases from no circulation at 1200 UTC 5 Sept. to $2.8 \times 10^6 m^2/s$ at 1200 UTC 7 Sept. The combined circulation is $3.4 \times 10^6 m^2/s$ at 1200 UTC 5 Sept. and increases to $5.8 \times 10^6 m^2/s$ by 1200 UTC 7 Sept. At the time of genesis, $52\%$ of the circulation of Nate comes from the region $R_1$ advected from Lee while the remainder is vorticity that was already present

---

[7]The non-advective isobaric absolute vorticity vector differs from the advective vector and therefore can cross material contours.

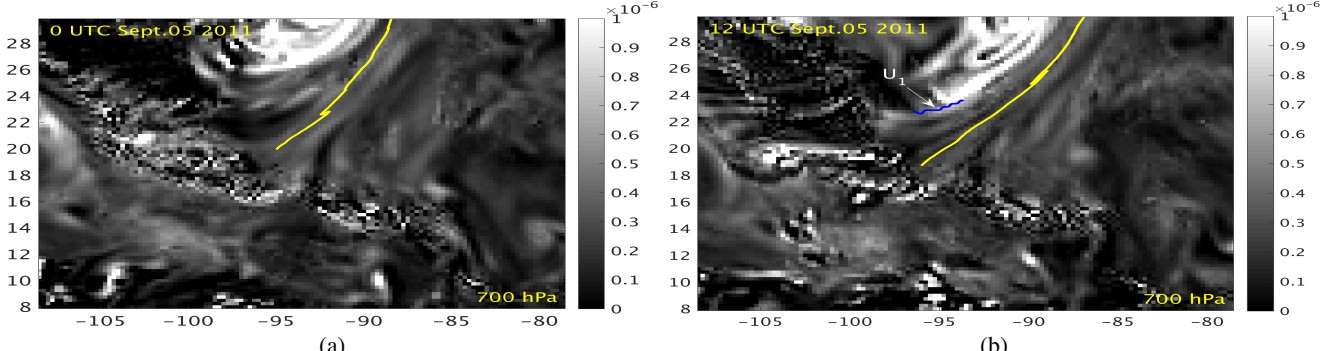

**Figure 2.** The unstable (yellow, blue) manifolds are overlaid on the ECMWF PV field $(K \cdot m^2/kg \cdot s)$ at 700 hPa in (a) and (b) at 0 UTC 5 Sept. and 1200 UTC 5 Sept.

in the area of low pressure in the Bay of Campeche. The mean vorticity in $R_1$ reaches $5.2 \times 10^{-5} s^{-1}$ at 0 UTC 6 Sept. before declining to $2.7 \times 10^{-5} s^{-1}$ at 1200 UTC 7 Sept. The mean vorticity of $R_2$ increases steadily to $2.8 \times 10^{-5} s^{-1}$ at 1200 UTC 7 Sept. In the ECMWF numerical data, the horizontal advection of vorticity accounts for over half of the change in circulation while the remainder of the change in circulation is due to nonadvective fluxes of vorticity. Similarly, the non-advective flux

contains unresolved advective fluxes. In principle, one might calculate the unresolved advective flux by subtracting the resolved non-advective flux from the residual of total tendency and advective flux.

The more complicated structure of the Lagrangian manifolds and their additional intersections allowed the formation of lobes. The lobe $L_1 = U_1(I_1, I_2) \cup S_2(I_2, I_1)$ is enclosed by $U_1$ and $S_2$, and is initially located to the north of Nate in the region of very dry air. Its advection can be seen in Figure 3 (b-g). Prior to development, the lobes did not penetrate into the center where

regions of highest OW were located, Figure 4 (c). The intersection points $I_1$ and $I_2$ travel cyclonically around the boundary and by 0 UTC 9 Sept. the intersection points have traveled close to $H_2$, as $L_1$ has begun to be ingested into Nate. During the entrainment of $L_1$, the very small lobe $L_2 = U_1(I_2, I_3) \cup S_2(I_3, I_2)$ is expelled from the vortex. At this time, the northern boundary of the pouch is redefined from $U_1(H_1, I_1) \cup S_2(I_1, H_2)$ to $U_1(H_1, I_3) \cup S_2(I_3, H_2)$, which reflects the inclusion of the contents of $L_1$ into the pouch interior and expulsion of $L_2$. $L_1$ originated from the dry air region and transported the dry

air toward the center. Though the development of Nate occurred on Sept. 7 before the dry air could reach the center, the dry air also contained lower vorticity than the moist air in the pouch, and reduced the mean vorticity within the pouch. By 0 UTC Sept. 8, $L_1$ had a negative relative circulation. Without the contents of $L_1$, the mean vorticity in the pouch was $3.1 \times 10^{-5}$, versus $1.8 \times 10^{-5}$ when $L_1$ is included.

In addition to $L_1$ and $L_2$, there are additional lobes, $L_3 = U_2(I_4, I_5) \cup S_1(I_5, I_4)$ and $L_4 = U_2(I_5, I_6) \cup S_1(I_6, I_5)$, labeled

in Figure 3 (b), that transport air across the southern pouch boundary. $L_3$, the interior of the region bounded by the cyan and magenta curves, contains moist air with high vorticity on the southern boundary of Nate, and travels inward transporting this air into Nate. $L_4$ contains a small amount of moist air with high vorticity that is initially in the interior of Nate, and over the three day time period, this air is expelled to the east through lobe transport. As the four lobes are transported, a rearrangement

of the boundary occurs, and by 1800 UTC 8 Sept., Nate contains the contents of the pouch from 0 UTC 6 Sept. minus the contents of $L_2$ and $L_4$, but with the addition of $L_1$ and $L_3$, Figure 3 (g). Since $L_2$ and $L_4$ are relatively small, they have little effect on the circulation.

**b. Relation of Lagrangian boundaries to tracers and convection**

The Lagrangian boundaries are closely related to the transport of tracers as Lapeyre et al. (1999) found that the maximum tracer gradient tends to align with the unstable manifold. The relationship of the manifold boundaries to both physical, e.g. potential vorticity and ozone ($O_3$), and advected tracers can be seen in Figure 4. The potential vorticity and $O_3$ fields are shown at 700 hPa at 1800 UTC Sept. 6 in Figure 4 (b) and Figure 4 (e), respectively. The intrusion of dry air entering the northwest side of the pouch is visible in the $\theta_e$ and $O_3$ fields, with higher $O_3$ and lower PV air contained in $L_1$. The gradient of the $\theta_e$

field, Figure 3, indicates a strong frontal boundary between moist and dry air to the north of the pouch that aligns very closely with $U_1$. However, the ozone field acts as a better tracer than $\theta_e$ since its filaments more closely follow the filaments of the manifolds, and isolines of $O_3$, Figure 4 (e), still approximately separate the air with origins from Lee (see $R_1$ in Figure 3 (a)) from the air with southern Gulf of Mexico origins. Small differences between the tracer gradients and manifolds are due to the non-conservation of the tracers.

Advected tracers form steep gradients purely from advective transport and can be seen by plotting the initial value of the advected quantity at the final location of the particle. Behavior similar to that of the other physical tracers can be visualized by the latitude tracer field (conserving initial latitude along trajectories), which shows the initial latitude of particles (Figure 4 (a)). In each case, there is an obvious alignment of the unstable manifold with the sharpest gradient of the tracer field.

We examine now whether the accumulation of moisture and confluent flow along the unstable manifold impacts the location

of convection. The 700 hPa stable and unstable manifolds are overlaid on GOES shortwave infrared 3.9 $\mu m$ brightness temperature (K) averaged over a 6 hour time interval spanning 0 UTC in Figure 4 (d). By 1800 UTC Sept. 6 a significant amount of moderately cold clouds are evident along the frontal boundary south of the unstable manifold ($U_1$, blue) in the southwest quadrant of the storm. The manifold boundary clearly partitions the cloudy region from the less cloudy regions. Though the moisture gradients align with the manifold boundaries, as seen in the relative humidity field in Figure 4 (f), the azimuthal

location of convection in relation to these boundaries is far less predictable, though it does tend to be on the interior of the boundary.

**c. Lobe transport in the WRF model**

The WRF model simulation is used to compare how temporal resolution, spatial resolution, divergence, and varying versus constant SSTs affect the manifold topology. Lobe transport is shown for the complimentary WRF simulation in Figure 5 where

the Lagrangian manifolds are overlaid on the vorticity field at the model $\eta$-level of 0.7, or approximately 700 hPa. Though the fine structure is different, the topology that emerges from ECMWF analyses and WRF outputs are very similar on the northern side of the disturbance with a pair of hyperbolic trajectories. A single large lobe is present in the dry air region to the north, with similar structure and location as $L_1$ from the ECMWF model, while a smaller filamentary lobe begins in the pouch and

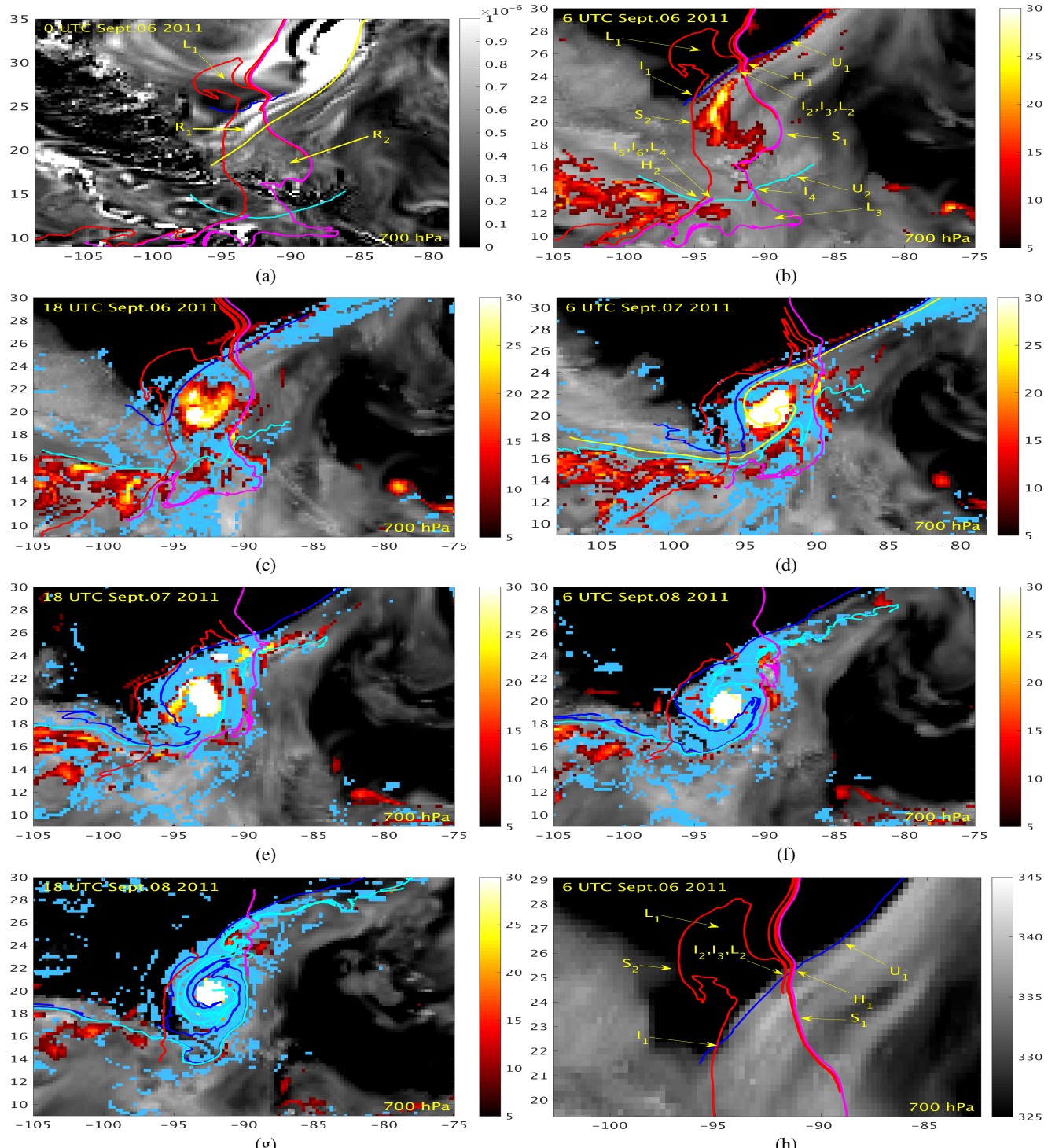

**Figure 3.** The stable (red, magenta) and unstable (blue, cyan) manifolds are overlaid on the ECMWF PV field $(K \cdot m^2/kg \cdot s)$ in (a) and $\theta_e$ field (K) (background with gray colors) at 700 hPa in (b-g) showing the time-evolution of the manifolds from Sept. 6 to Sept. 8. The hot colormap shows the $\zeta_{Lag}$ values while the cool colors indicate $OW_{Lag}$ values of less than -8 indicating high strain. Labels are provided in panels (a) and (b). A zoom of the features that are shown in panel (b) is shown in panel (h), which also shows the colorbar for the $\theta_e$ field. An attracting line from the Lee flow is shown in yellow in (a), while the advection of that line is shown in (d).

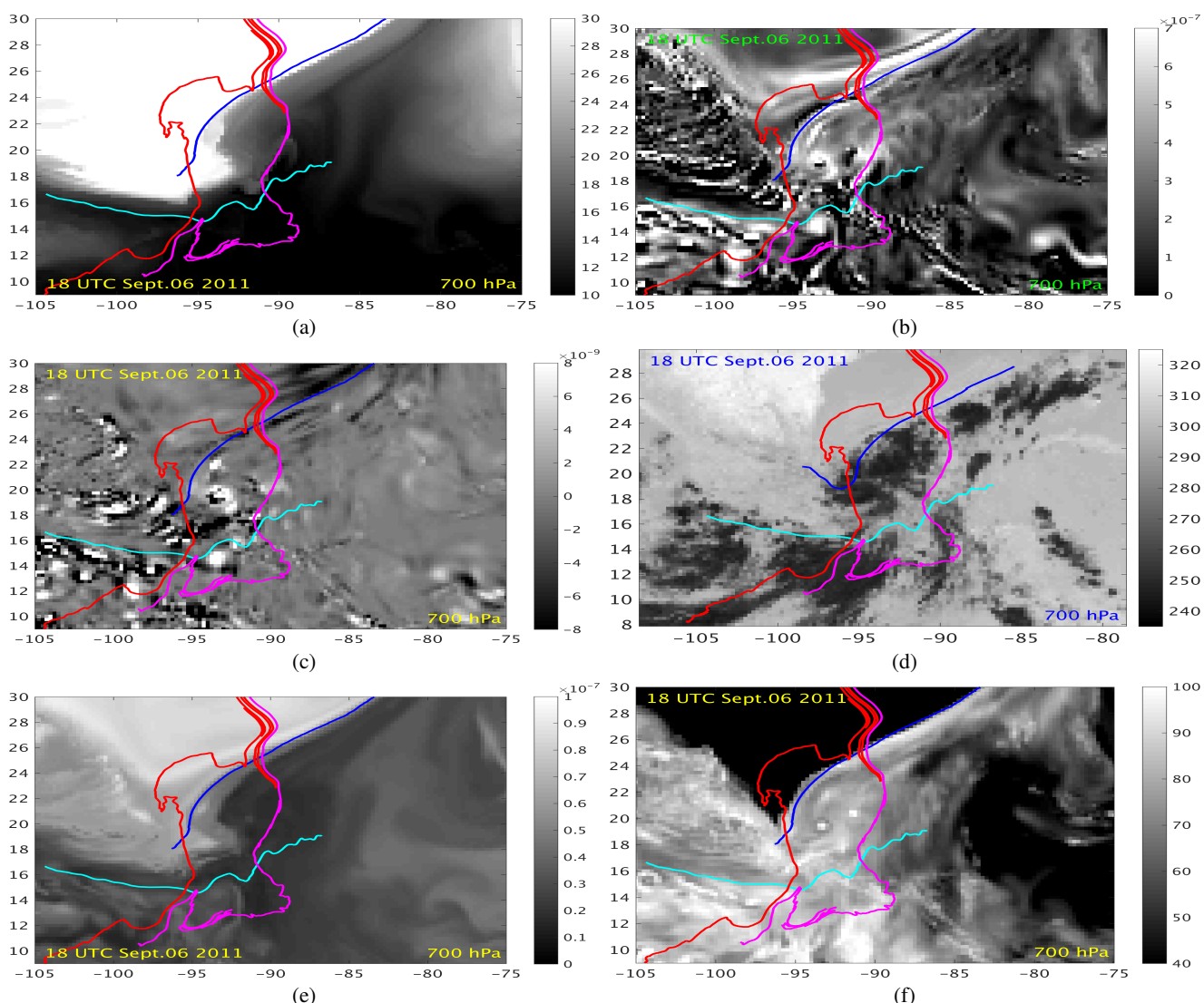

**Figure 4.** The stable (red, magenta) and unstable (blue, cyan) manifolds are overlaid on the latitude tracer field (degrees) in (a), the PV field $(K \cdot m^2/kg \cdot s)$ in (b), the $OW$ field $(s^{-2})$ in (c), the GOES 3.9 $\mu m$ brightness temperature (K) in (d), the ozone mixing ratio (g/kg) in (e), and the relative humidity field (f) at 1800 UTC Sept. 6 at 700 hPa.

is ejected. Similar lobe transport is present in the WRF simulation on the southern side as in the ECMWF analyses. As noted earlier, the ECMWF analyses at 6 hour time intervals is not sufficient to close the circulation budget over a 72 hour time interval as a large residual non-advective flux term remains. The advantage of the finer temporal resolution is that the circulation budget has advective and tilting fluxes that remain larger than residual non-advective fluxes for a materially advected region. A detailed circulation analysis of the non-advective fluxes for the WRF output is shown in Rutherford and Dunkerton (2017).

### d. Lobes on isentropic surfaces

Due to the existence of the frontal boundary and large moisture and temperature gradient on the north side of the pouch, one may question whether manifolds computed with isobaric velocities represent realistic particle motions. The manifolds using isentropic velocities at the 315 K level, representative of the $\eta = 0.7$ level potential temperature during the genesis time period, are overlaid on the isentropic vorticity in Figure 6 (a)-(b). While some of the details of the manifolds change from the $\eta = 0.7$ level, the separatrix remains similar, with lobes $L_1$ from the dry region and $L_3$ from the southern moist region responsible for most of the transport from the environment into the pouch. The agreement of isentropic and isobaric manifold analyses is consistent with a hypothesis that the baroclinic characteristics of the frontal zone do not penetrate significantly into the immediate environment of Nate, that is, the inner pouch region. Such may not be the case for tropical storm formation in general, e.g., in tropical transition from a baroclinic precursor. However, the advantage of isentropic analysis is lost when parcel motions are no longer adiabatic on short time scales, as in regions of intermittent deep convection. We prefer isobaric analysis for this reason, irrespective of sloping isentropes, and for a more general reason that isobaric surfaces remain stratified (monotonic in height) everywhere in the atmosphere. Isentropic surfaces, on the other hand, are non-monotonic in breaking gravity waves and ill-defined in moist buoyant fields (convective clouds) and neutrally stratified dry boundary layer. Outcropping of isobaric surfaces in high topography or intense low pressure (e.g., hurricane) might be avoided with a sigma or hybrid coordinate, if desired, but it may be equally desirable to retain the isobaric formalism and to calculate pressure and frictional torques explicitly, together with trace constituent sources and sinks, along such physically constrained manifold boundaries. Issues associated with topography are thought to be second order effects in this study and consequently lie outside the scope of the present study.

### e. Non-divergent lobe transport

Using the full flow field on constant $\eta$-surfaces, convergence leads to a net entrainment of vorticity contained in lobes, and it is not surprising that the area of the entrained lobes is far greater than the area of the expelled lobes. To further examine how the lobes are created, we again locate the manifolds using the non-divergent flow, computed using a Helmholtz decomposition on the WRF velocity fields. The manifolds are overlaid on the water vapor mixing ratio in Figure 6 (c)-(d). The hyperbolic trajectories are very similar to those in the divergent flow. The lobes on the north side of the pouch are still present, and result from velocity fluctuations along the frontal boundary, though their size is much smaller and these lobes do not become entrained into the core, but are instead transported parallel to contours of water vapor at the edge of the pouch. The time-variation can also be seen by the folding of the unstable manifold near the western hyperbolic trajectory. However, the manifold configuration to the south of the pouch has changed as there are no longer lobes; in fact the time evolution of the unstable manifold shows a

very slow time variation. While time-dependence of the rotational flow alone is sufficient for lobe transport along the frontal boundary, it is time-dependence of the divergent flow that is responsible for lobe transport to the south. This observation indicates that lobe transport is tied to convection and the fact that there are a pair of lobes entrained over approximately one day in the divergent flow leads us to question whether 2D lobe transport could be a response to the diurnal cycle of convection.

## f. Effects of varying SSTs

The role of varying SSTs is investigated by an additional WRF simulation using temporally constant SSTs at the initial time of 0 UTC 6 Sept so that the upwelling that occurred from Nate has no feedback into the simulation. The manifolds at $\eta = 0.7$ are overlaid on the vorticity field in Figure 6 (e)-(f). With constant SSTs, the disturbance intensifies slightly more quickly, reaching a maximum $\eta = .7$ vorticity of $9.35 \times 10^{-4}$ by 0 UTC 7 Sept. compared with a maximum of $8.75 \times 10^{-4}$ for the varying SST case. Though the fine scale structure is slightly different, the higher SSTs do not cause a topological change in the manifold configuration at $\eta = 0.7$. At other levels, e.g. $\eta = 0.5$ and $\eta = 0.6$ (not shown), the manifolds to the north remain the same, but the manifolds to the south do not have additional intersections allowing moist air with high vorticity to be entrained. Both the varying SST simulation and constant SST simulation show similar system-scale convergence at $\eta = .7$, as the area of the pouch is reduced to .72 times it original size, see Figure 7 (a). The circulation (Figure fig:circarea (a)) and mean vorticity (Figure fig:circarea (b)) are slightly higher for the constant SST simulation, indicating only a modest impact of the upwelling to storm intensity. Differences between the two simulations are similar at other levels as they are at $\eta = .7$.

The WRF simulations collectively demonstrate that varying SSTs and resolution have little impact on manifold structure or on the contribution of the manifolds to the circulation since the small filaments emanating from the lobes contain very little circulation. The primary Lagrangian structures, $L_1$ to the north, $L_2$ to the south, and the pouch, remain robust features that are relatively insensitive to fine-scale variations. Though the manifolds still produce the same number of intersections and lobes with non-divergent flow, the lobes are much smaller and are not entrained to the center. Analysis of the WRF simulations through the mid-troposphere (not shown) also supports these conclusions.

## g. Vortex radial structure

We now consider the radial profiles of important kinematic quantities including the strain rates, where radius is taken with respect to the best-track storm location. Both OW and the sum of the squares of strain rates are translation invariant quantities, so they do not depend on the choice of translating Eulerian reference frame, while the strain rates depend individually on the choice of coordinate system. We orient the coordinate system along the direction given by the tangent to particle motion by the transformed velocities $(\tilde{u}, \tilde{v})^T = T^{-1}\mathbf{u}$, where $T = \frac{1}{\|\mathbf{u}\|}[\mathbf{u}, \mathbf{n}]$ and $\mathbf{n} = (-v, u)$ is the outward normal vector. The velocity gradient tensor in this coordinate system is given by

$$\nabla \tilde{\mathbf{u}} = T^{-1}\nabla \mathbf{u} T = \begin{pmatrix} \tilde{u}_\parallel & \tilde{u}_\perp \\ \tilde{v}_\parallel & \tilde{v}_\perp \end{pmatrix} \tag{7}$$

The shear and normal strain rates are given by $S_n = \tilde{u}_\parallel - \tilde{v}_\perp$ and $S_s = \tilde{v}_\parallel + \tilde{u}_\perp$ respectively, and satisfy $S_n^2 + S_s^2 = S_1^2 + S_2^2$. In this rotated coordinate, the strain rates are oriented parallel to the tangent vector $(S_s)$ and normal to the tangent vector $(S_n)$.

The time evolution of radial profiles for OW, shear strain, relative humidity, and vorticity from the ECMWF data is summarized in Figure 8 (a)-(d), respectively. The line marking the radius of the shear sheath, where the radial average of OW becomes negative, is shown as a function of time in white. The $\zeta$ time sequence shows higher average values toward the pouch center as time increases. Likewise, OW shows an increase near the center. As the vorticity is enhanced by vertical mass flux near the center, the strain regions outside the regions of high vorticity converge inward toward the limit cycle, but remain just outside the region of highest vorticity. However, while $\zeta$ decreases slowly outside the center as the highest vorticity is concentrated in the core, the OW values show a dramatic decrease to negative values just outside the core due to higher strain. As the inward transport along the lobes progresses, the manifolds lengthen, increasing the area within the pouch that is dominated by strain. From Figure 8 (d), we see that as the unstable manifold is entrained near the center toward a small radius, elevated values of $S_s$ appear just outside the strongest rotation at the radius where the unstable manifold is entrained. High OW and $\zeta$ values are present inside this radius as the flow is close to solid-body rotation. As it is entrained, the strain along the unstable manifold changes from stretching normal to the manifold to parallel shearing along the manifold. This shear boundary protects the vortex from further interaction with low-vorticity dry air, and allows concentration of high vorticity air near the center of the pouch. Thus, high OW values at the pouch center cannot occur without high strain just outside the region of high OW, and this radial profile of strain versus rotation are the leading indicator of higher $\zeta$. The importance of the shear sheath can be seen in the relative humidity profile, Figure 8 (c), where radial averages of relative humidity are very high within the shear sheath, but the averages drop quickly outside the shear sheath.

**h. Backward trajectories**

Backward trajectories provide additional details about the impermeability of the inner core. Trajectories were seeded uniformly at 0 UTC 9 Sept. within the inner core boundary defined as a circle 1 degree of the storm center and advected backward in time isobarically to 1200 UTC 6 Sept. Their radius from the circulation center and $\theta_e$ values are plotted at 12 hour intervals in Figure 9 where the temporal positions are marked in different colors. The 500 hPa and 700 hPa backward trajectories are shown in panels (a) and (b), respectively, where orange dots inside a radius of 1 degree indicate that trajectories from the inner core on 0 UTC Sept. 9 were completely contained within the pouch boundary during the entire integration until 1200 UTC 6 Sept. None of these trajectories were significantly drier at earlier times, acquiring a $\theta_e$ increment of approximately 4-6 K at 500 hPa and 6-10 K at 700 hPa. During this gradual moistening trend, air is brought from over 3 degrees, but still within the pouch, to smaller radius in the core.

The 850 hPa and 925 hPa trajectories are shown in (c) and (d), and indicate that a portion of the trajectories originated from much drier regions, and moistened significantly during their entrainment. Many of these trajectories originated at locations further than 8 degrees from the center and their $\theta_e$ values increase by as much as 40 K coming inward toward center. Tracking the initial latitude of these trajectories (not shown) indicates that almost all of them came from north of the pouch, consistent with what is already known from the manifold locations. The moistening in the boundary layer associated with surface moisture

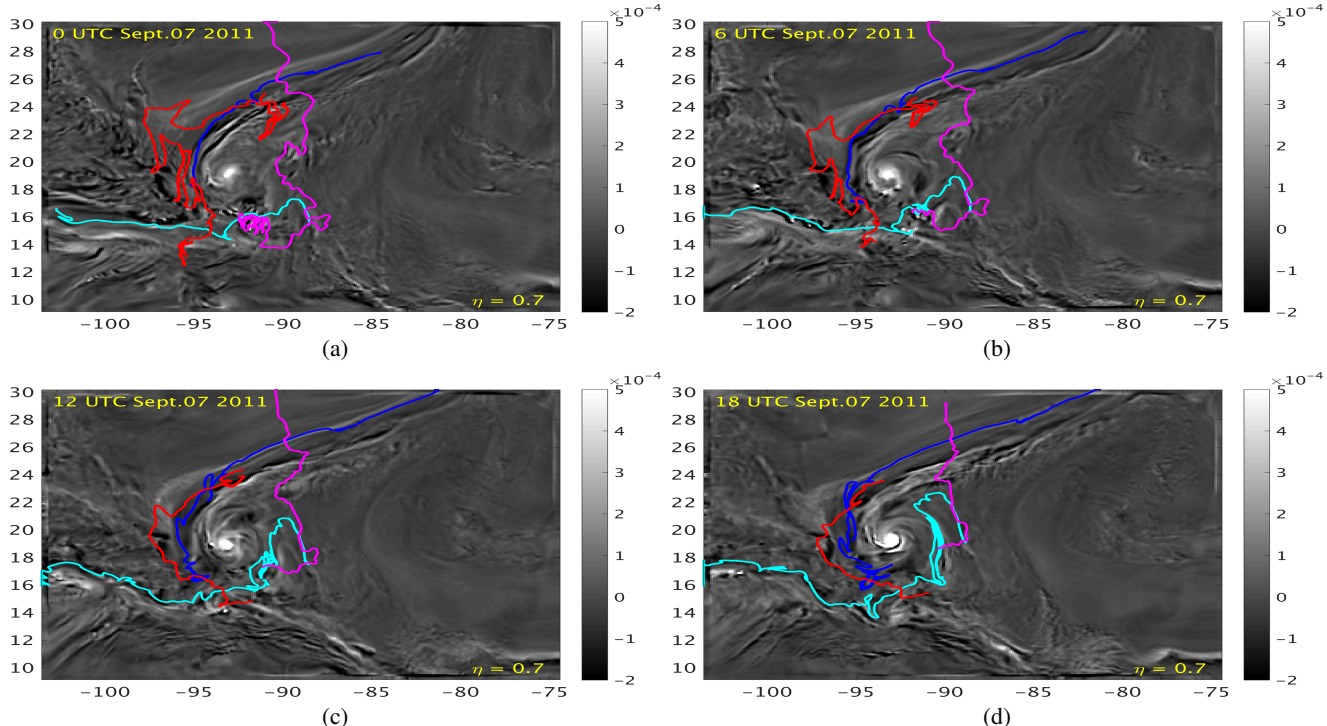

**Figure 5.** The stable (magenta, red) and unstable (blue, cyan) manifolds from the WRF simulation using non-uniform SSTs are overlaid on the vertical vorticity $(s^{-1})$ in (a-d).

fluxes over sea and entrainment from low-level convergence are not surprising. However, we see that a portion of the 850 hPa and 925 hPa trajectories are within 2 degrees of the center with $\theta_e$ values less than 330 K at 0 UTC 8 Sept., just prior to the period of weakening. At this time, Nate was over cooler water, and the entrained air was not moistened.

An idealized study by Braun et al. (2012) found that disturbances that reside very close to dry air and entrain dry air to within approximately 200 km of the circulation center may develop more slowly but achieve a similar maximum intensity. Our findings here indicate that the radial distance of entrainment from the center may not be the only factor limiting development, but whether the intrusion is able to penetrate the shear sheath. However, even without complete penetration, dry air may influence the storm by modifying the inflow layer as shown by Powell (1990) and Riemer and Montgomery (2011). An additional modeling study showing the different depths of entrainment of manifolds would be required to completely understand the implications of entrainment depth on development. Yablonsky and Ginis (2008) and Yablonsky and Ginis (2009) showed that oceanic upwelling may be a limiting factor in storm intensity also, and we suggest that it may have been a factor in Nate's weakening prior to landfall.

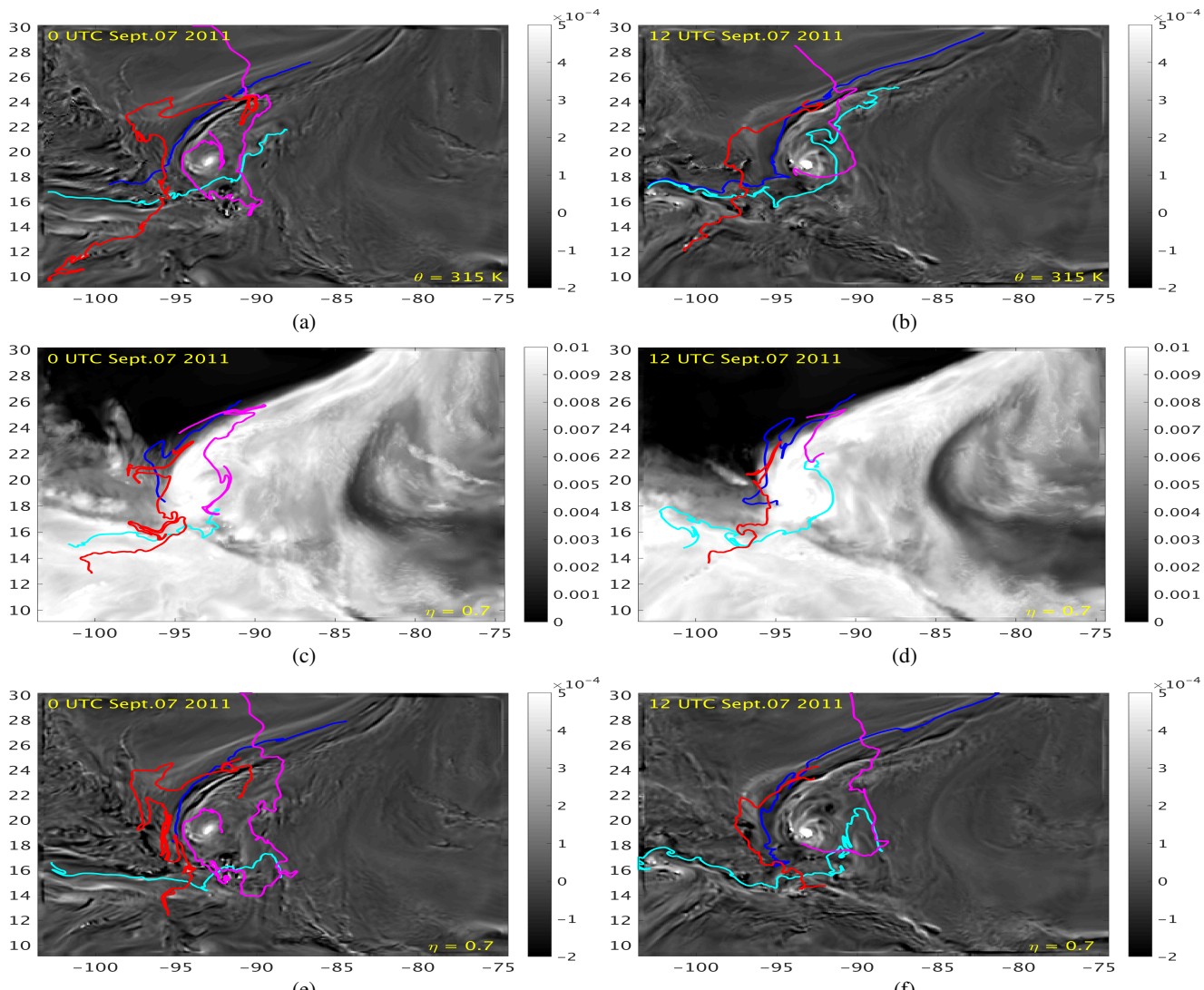

**Figure 6.** The stable (magenta, red) and unstable (blue, cyan) manifolds from the WRF simulation at the $\theta = 315$ K level using varying SSTs are overlaid on the vertical vorticity ($s^{-1}$) in (a-b), manifolds from the non-divergent velocity field from the WRF varying SST simulation are overlaid on the water vapor mixing ratio ($g/kg$) in (c-d), and manifolds at the $\eta = 0.7$ level using constant SSTs are overlaid on the vertical vorticity ($s^{-1}$) in (e-f).

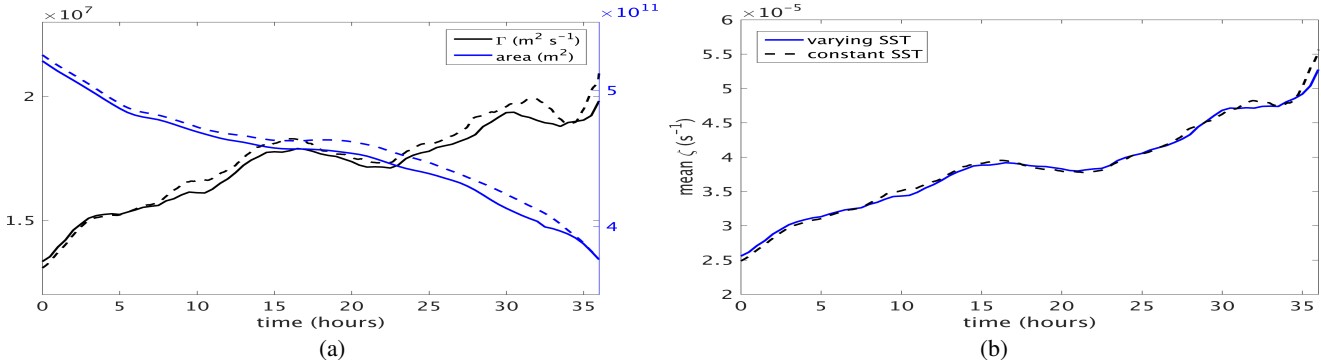

(a)                                                                                      (b)

**Figure 7.** The circulation $(m^2 s^{-1})$ and area $(m^2)$ are shown in (a) as black and blue curves resp. for the varying SST (solid lines) and constant SST (dashed lines) WRF simulations, respectively. The mean vorticity $(s^{-1})$ is shown in (b) for the varying SST simulation (solid) and constant SST simulation (dashed).

**OW**

**Shear strain**

(a)                                                                                      (b)

**Relative humidity**

$\zeta$

(c)                                                                                      (d)

**Figure 8.** The radial (degrees) profiles of OW $(s^{-2})$, shear strain $(s^{-1})$, relative humidity $(\%)$, and vorticity $(s^{-1})$ from the ECMWF data are shown in (a)-(d), respectively, from Sept. 0600 to 1800 UTC Sept. 9. The radial location of the shear sheath is shown in white.

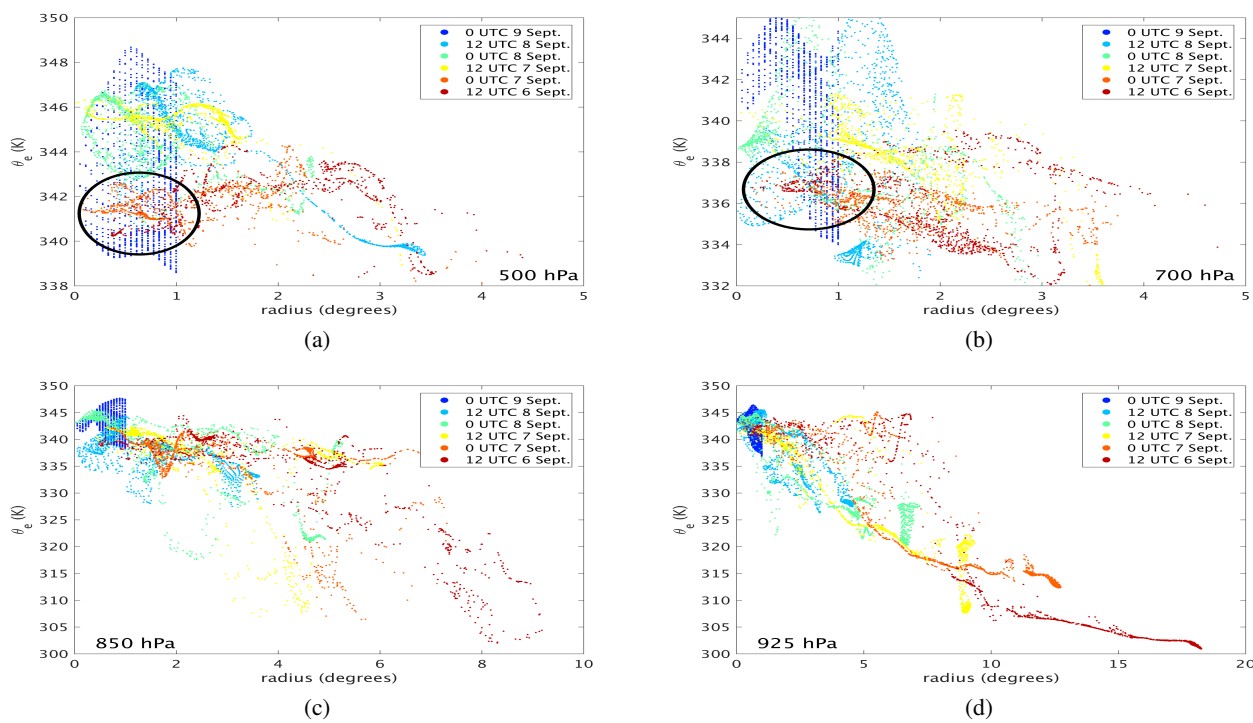

**Figure 9.** The values of $\theta_e$ versus radius from circulation center are plotted for sets of isobaric trajectories from the ECMWF data released within the inner core boundary at 0 UTC 9 Sept. and integrated backward to 1200 UTC 6 Sept. at 500 hPa, 700 hPa, 850 hPa, and 925 hPa in (a)-(d) resp. The different colors indicate the time at which the properties of the trajectories were analyzed. Sets of trajectories that remain in the core during the entire integration are located inside the black ellipse in (a) and (b).

## i. Horizontal transport at 500 hPa and 850 hPa

We now examine the vertical structure of the pouch by identifying the manifold structure on other levels. The manifolds are shown at 18 UTC Sept. 6 to 0600 UTC Sept. 8 at 850 hPa (left column) and 500 hPa (right column) in Figure 10. These manifolds were identified by the same methods as those at 700 hPa, and the unstable manifolds show a configuration very similar to those at 700 hPa. The stable manifolds at the other levels have some important differences from those at 700 hPa.

At 500 hPa, the structure is very similar to the structure at 700 hPa, where the manifolds form a complete pouch boundary, and allow only a small intrusion of dry air from the north that is contained within a lobe. A very similar structure (not shown) can be observed at 400 hPa and 600 hPa, though it does not extend above 400 hPa.

At 850 hPa, the manifold structure differs from those found from 700 hPa to 500 hPa in that the stable manifolds do not have additional intersections with the unstable manifolds other than at the locations of hyperbolic trajectories. As the manifolds evolve, the northern unstable manifold is entrained inward, leaving a large region of dry air that can enter the vortex. Lobe

transport does not apply here and entrainment of dry air is not limited to the contents of the lobes, but rather, the total flux through the open pathway.

At both 500 hPa and 850 hPa, unstable manifolds are entrained into a limit cycle of circular flow with no further entrainment, and like at 700 hPa, the change of hyperbolic to shear stability of the manifold forms a shear sheath that provides some protection to the inner core, defined kinematically as the region with strong recirculation seen, e.g., by large positive OW values, from the intruding dry air, Figure 10 (c)-(h). However, at 850 hPa, some dry air has been entrained into the inner core prior to the formation of the shear sheath at 1200 UTC 9 Sept., Figure 10 (c) and (e) (see also Figure 9 (c)). At 500 hPa, the shear sheath, seen by the limit cycle (cyan curve) at 0 UTC 9 Sept., is established prior to the entrainment of dry air.

## 5 Conclusions

In this paper, we have explored how the rearrangement of Lagrangian flow boundaries may limit the dry air in the vicinity of a tropical disturbance from interacting with the disturbance. Hurricane Nate developed despite a region of very dry air in close proximity to the storm. While the storm-relative frame showed closed streamlines, the stable and unstable manifolds defined invariant regions called lobes that can transport intruding dry air into the pouch toward the storm center, but failed to penetrate the core of the proto-vortex. A shear sheath served to protect the center by maintaining a strongly deforming radial shear which, in turn, allowed vorticity concentration of the core to continue, undiluted with dry air and weaker vorticity.

We offered a dynamic view of the pouch for Nate that describes the entire storm evolution at fixed vertical (e.g. isobaric) levels based on the evolution of Lagrangian flow boundaries. In this view, we found that the Lagrangian pouch structure showed the sources of air that were advected into the pouch. We also showed that the advective fluxes of vorticity into the pouch can be measured by lobe transport, and account for over half of the vorticity that Nate acquired. The transport that we see in this case is consistent with the radial profiles which showed the accumulation of the manifolds, an increasing tracer gradient, and a shear sheath that marks an additional transport boundary to the inner core.

When Nate was a tropical storm in close proximity to dry air, the entrainment of dry air was limited to what was transported by lobes when an enclosed Lagrangian boundary was present.

The Lagrangian boundaries lead us to a material description of the transition from a large-scale pouch boundary that blocks environmental dry air during genesis to a much smaller vortex core that is present after genesis:

1. The frontal boundary rolls up and combines two air masses, one from each side of the frontal boundary.

2. During the rollup, hyperbolic trajectories become detectable along the boundary, indicating the existence of stable and unstable manifolds.

3. The folding of manifolds near the hyperbolic trajectories implies the existence of additional intersection points of the manifolds that are not the hyperbolic trajectories, but these intersection points travel cyclonically toward the hyperbolic trajectory. The manifold segments between adjacent intersection points mark the lobe boundaries.

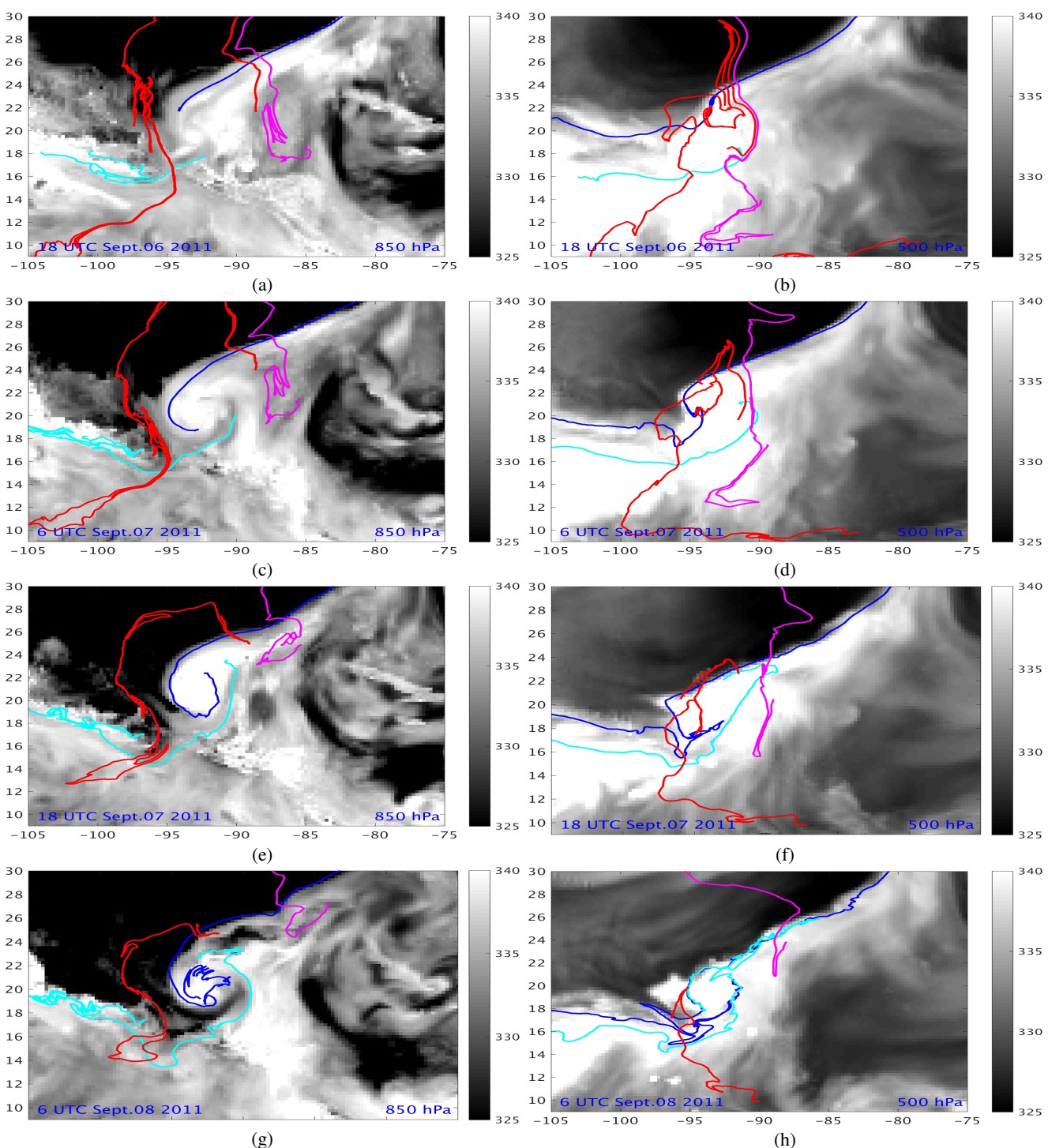

**Figure 10.** Stable (red, magenta) and unstable (blue, cyan) manifolds are overlaid on the ECMWF $\theta_e$ field (K) and show the pouch boundary from 1800 UTC Sept. 6 to 0600 UTC Sept. 8 at 850 hPa (left) and 500 (right) hPa.

4. Within the pouch, wrapping of the unstable manifold reaches a limit cycle surrounding the inner core as convergence concentrates vorticity from the moist region into a vortex core.

5. In a competing process, additional entrainment of lobes allows the import of dry air towards, but not necessarily into, the core.

6. As vorticity is concentrated into the core, the unstable manifold lengthens, and the elongated manifolds and lobes form a shear sheath barrier to transport of additional dry air into the core[8].

There are two configurations of manifolds that describe the transport of dry air toward the storm center along the manifolds. At 700 hPa and 500 hPa, lobe transport and a rearrangement of a separatrix allowed a region of dry air to enter the pouch. However, the dry air region was contained and did not penetrate the inner core boundary due to the presence of the shear sheath. At 850 hPa, the manifolds showed that there was a direct pathway of transport into the pouch that still reached a limit cycle before reaching the circulation center, and the pathway was wider than suggested by translating Eulerian streamlines. These two mechanisms for dry air transport compete with the aggregation of cyclonic vorticity. Lobe transport is a slower process which limits the amount of dry air entering the vortex, while an open pathway allows continual entrainment of dry air.

Based on companion numerical integrations of this case, the advection of manifolds is somewhat sensitive to SST, pressure level, integration time and numerical model (details). While the fine-scale details of the manifolds may differ considerably, the differences between manifolds computed in different ways are confined to filaments that have little circulation and are homogenized. The conclusion, that dry air from the north of Nate entered in and corresponded to the transport of one lobe, while moist air that entered Nate was confined to another lobe, is robust.

Further study on the rate of entrainment versus the rate of vorticity aggregation in an idealized setting and in modeling studies will help clarify the role of dry air intrusions that are partially ingested into developing storms in slowing but still allowing development. The techniques used here should be useful for those studies.

**Acknowledgments**

The authors would like to acknowledge NSF grants AGS-1313948, AGS-1439283, AGS-0733380 (now expired), and NASA grant NNG11PK021. The authors would also like to thank Drs. Gerard Kilroy and Roger Smith and the German Weather Service for providing us with the ECMWF global model data for this basic research investigation.

---

[8]In this case, entrainment of manifolds forms a transport barrier at the edge of the inner core where the entrainment of lobes is due to time-dependence of velocities. The end result is similar to the effect of divergence in steady flow studied in Riemer and Montgomery (2011). Alternatively, Rutherford et al. (2012) found that the remnant manifolds exterior to VHT's may help to form the boundary without being attached to a hyperbolic trajectory at the edge of the pouch. These processes are not mutually exclusive and the shear sheath is a combination of these processes.

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
