# Peer review of "The genesis of Hurricane Nate and its interaction with a nearby environment of very dry air"

_Atmospheric Chemistry and Physics, 2016_

## Referee Comment (RC1) · D. Ahijevych (Referee) · 15 Mar 2017

General Comments

This paper extends the marsupial paradigm of tropical cyclogenesis to a non-African Easterly Wave. Without a wave-centric reference frame, air parcels around the storm are divided into two camps: those inside the pouch and outside. It is shown that some mixing occurs across the pouch boundary due to lobes breaking off, but dry air is shielded from the inner vortex by a shearing sheath that can be traced back to the manifolds that separate pouch and non-pouch air.

I almost cut my review short because I had never heard of the word "manifold" outside of a car engine setting, but after reading through the paper, which is fairly well written, I now have a decent idea of what it means. Now that I've read the paper, the abstract

and introduction make much more sense.

Later in the paper there is a weaker section with some mistakes in the figure and text, but overall this is ready for publication. My technical corrections are the main component of this review.

Some of the paper is still way beyond my area of expertise, and I have no idea whether it is right or wrong, such as the section on "Manifold computations" (pages 7-8).

Specific comments

After rereading the abstract it isn't clear why source the 'tilting' mechanism is given such prominence. It is barely mentioned again in the paper.

The adjective "layer-wise" used in the abstract and other places is confusing. Could you replace "layer-wise advection"? with "advection on a constant pressure level"?

Page 3, line 5. It isn't clear what "non-advective fluxes" are. This becomes apparent later on, but it is first mentioned here without any explanation.

Page 9, line 15. It is not obvious that the pouch region that originated from Lee still contains high potential vorticity air. From the figure, PV is muddled and similar to the PV in the portion with Gulf origins.

Technical corrections

Page 3, footnote 2. Change "if" to "of".

Page 7, Figure 1 caption. Change 'L2' to 'L4'.

Page 7, line 9. What is an "LCS"?

Page 7, line 10. What is "FTLE"?

Page 8, line 29. U2 is never labeled in the figure. That would be nice.

Page 9, line 2, 3rd-to-last word. Change "northeast" to "southwest".

Page 9, line 6, last term should be S1(I4,H1), not S2(I4,H1).

Page 9, footnote 7. What is "VS"?

Page 10, line 19. Can the lobe labels of L3 and L4 in Figure 2 (a) be reversed so as to match the order of the labels in FIg. 1? In other words, change L3 to L4 and change L4 to L3. ?

Page 10, line 25. Explicitly reference Fig. 2 after mentioning the gradient of theta-e. I was incorrectly looking for it (theta-e) in Fig. 3.

Page 10, line 26. "north" should not be capitalized.

Page 12, Figure 3 caption. Is the satellite panel described accurately? The text says it is 0.6 um, but isn't that visible? Wouldn't that be white for clouds? Not sure what the units are of the color table and it isn't clear how it could be 0.6 um imagery.

Page 13, Line 21. Manifolds are overlaid on water vapor not vorticity in Figure 5.

Page 14, first paragraph. Fig doesn't seem to go with text. Text talks about 700 mb vorticity and area. Figure has circulation and mean vorticity. Figure would also benefit from un-squished text. Y-axis ranges of panel (a) should be same as panel (b).

Page 14, line 18. Panel labels don't start with (a) in text reference or actual Figure 8.

Page 14, last paragraph. This paragraph is confusing. In line 21, when you say "as the vorticity moves inward", do you mean the vorticity maximum (in the radial profile) is moving inward? And in line 27 what is meant by "the stability of the unstable manifold"?

Fig 10, and Page 15, line 28. I think this time range is Sep 6 to 8, but can't read the time labels on some panels.

Page 16, line 6. THe term "limit cycle" is first used. I was familiar with it, but I understand it now after reading further.

Page 16, lines 8-10. Something with wrong with references to Fig. 9 (c)-(h), which

don't exist.

Page 20, FIg. 8. Need units in figure or caption.

---

## Referee Comment (RC2) · Anonymous Referee #2 · 21 Mar 2017

The authors present an analysis of the formation of Tropical Cyclone Nate (2011) from the perspective of hyperbolic Lagrangian boundaries. In addition, they discuss the role of a so-called shear sheath as a further transport barrier. The authors find that, between 700 and 400 hPa, Nate developed within a "pouch", i.e. a region that is closed with respect to horizontal transport. The Lagrangian boundaries (the "pouch") protected the developing system from a nearby region of very dry air. Entrainment of air from outside into the pouch by so-called lobe dynamics is given special attention in this manuscript. At 850 hPa, however, the "pouch" was not closed and thus an open pathway existed through which environmental air could be transported towards the core.

General comments:

[Figure]

The authors follow up on their previous work and continue to investigate tropical cyclogenesis from the innovative and promising perspective of Lagrangian transport barriers. Overall, this manuscript contains much interesting material and will eventually shed important insight into the development of Nate (2011). However, while the introduction and the conclusions are well written, the presentation of the results, and partly that of the methodology also, requires improvement. Furthermore, there is too much discussion of aspects that are not sufficiently introduced or shown in the manuscript. A revised version should have a stronger focus on the material that is actually presented, or should extend the material to better support some of the discussion/ statements. I have conceptual concerns with this manuscript also, specifically with i) the 2D Lagrangian analysis on pressure levels in the vicinity of a baroclinic zone, ii) the emphasis of hyperbolic manifolds vs. elliptic Lagrangian structures, and iii) usage of ECMWF data vs. higher (temporal)-resolution WRF data (see specific comments below). Overall, major revisions are required before this manuscript can be published in final form.

Specific comments:

* Validity of 2D analysis: The manuscript provides little motivation why analysis of Lagrangian boundaries of the quasi-horizontal flow should provide insight into the development of Nate (2011). Certainly, the authors make some assumptions that they had articulated in earlier work. These assumptions, however, should be stated also in this manuscript. Importantly, the 2D assumption seems to be in contrast with a statement made by the authors about the role of convection in lobe transport (pg 13, line 28). Clarification is required. Nate develops near the boundary between two air masses. The authors emphasize the large moisture gradient across this boundary. Arguably, the very dry air to the North of Nate is of midlatitude origin and presumably this air mass is also considerably colder than the tropical air mass in which Nate develops. In short, I expect a strong baroclinic zone to the North of Nate and the large-scale, 2D flow follows isentropic rather than isobaric surfaces. I question that the analysis on

isobaric surfaces is indeed Lagrangian, in the sense that the authors follow air parcels transported by the large-scale (adiabatic) flow, which is most likely one of the non-articulated assumptions made by the authors. Convincing justification for the use of an isobaric framework in the presence of strong baroclinicity is needed.

* The authors focus on the objective identification of hyperbolic Lagrangian structures. Similar methods can be applied to identify elliptical Lagrangian structures, which play an important role in separating a vortex from its environment also. The authors appreciate this role by their qualitative discussion of the "shear sheet", e.g. on page 6. For a wave's critical layer (e.g. Dunkerton et al. 2009) there is conceptual understanding why the flow boundaries that arise from the environmental flow – and are thus the relevant boundaries, in which the embryo tropical cyclone develops – are hyperbolic structures. Such conceptual background misses for non-AEWs disturbances like pre-Nate, or at least the authors do not provide such background. Therefore, an objective identification of elliptical boundaries will considerably strengthen this manuscript. In addition, the identification of elliptic boundaries would help to introduce the concept of a limit cycle, which is referred to later in the manuscript, and help to define the core of the disturbance, which is undefined in the current version of the manuscript.

pg 1, line 15; vorticity generation by tilting: This aspect is hardly touched on in this manuscript. I recommend omitting reference to this process in the abstract (and in the conclusions).

pg 5, line 1; and elsewhere; "Eulerian boundary": There are references in the manuscript to Eulerian streamline patterns that are not illustrated in this manuscript. In addition, there are references to the role of tropical cyclone Lee that are not illustrated either. For the reader, it is rather hard to follow (and appreciate) these descriptions. I suggest using one or two additional figures to illustrate such points; or to keep such references to a minimum.

pg 7, The subsection "Manifold computations" requires considerable improvements:

The authors use phrases like "some situations" and "additional options" but it remains unclear if or when other options are used or what methods are applied in other situations. Most importantly, it remains unclear from this description for how long the underlying trajectories have been calculated. It is well known that finite-time Lagrangian coherent structures are sensitive to the integration time. A more explicit discussion of this integration time and a discussion of the sensitivity of the results to integration time are needed.

pg 10, "Relation of Lagrangian . . .": Unfortunately, the presentation of the results deteriorates rather significantly from here on. E.g., the authors note that PV and O3 is shown and then continue with a discussion of theta_e, the GOES imagery is presented without units, convergence is presumably confused with confluence (pg. 13, line 1), results from WRF at 600 mb (which should read hPa) are compared to results from ECMWF data at 700 hPa, vorticity is confused with mixing ration in Fig. 5, it is unclear what the difference is between individual panels in Figs. 4-6, . . . The subsection "Backward trajectories" is very dense and it seems as some important information is not given to the reader. I cannot identify in the figures several features described by the authors. This is of particular importance with respect to the vertical similarity of manifolds and the limit cycle.

* The comparison between the ECMWF and the WRF data is confusing. Importantly, it is not clear how much the results based on the ECMWF data can be trusted. Furthermore, the comparison of the results using the full wind field and the non-divergent flow only is poorly motivated.

* The conclusions refer to several aspects that have not been discussed sufficiently in the manuscript. The arguments given in enumerations 1) and 2) are plausible but have not been shown in this manuscript. The "core" referred to in enumeration 5) has never been defined. Finally, the Eulerian streamlines noted on pg 19 have not been shown in this manuscript. The revised conclusions should focus much more on results and insight that is shown and developed in the manuscript at hand.

Technical corrections/ Editorial recommendations:

pg 1, line 2: of –> or

pg 2, line 33; kinematic structures as consequence of invertibility of vorticity: I cannot follow this argument. Please clarify.

pg 3, line 2; "arm of T.S. Lee": incomprehensible

pg 3, line 5: should read "for a non-AEW disturbance"

pg 2, line 12; suggest: initiated – > first identified

pg 5: should "vortex strip" read "vorticity strip"?

pg 5, line 20: according to the references it should read "Rutherford and Dunkerton, 2017"

pg 6, line 4: This sentence seems to lack something, maybe "vorticity" after "system"?

pg 6, line 5; "isobaric vorticity substance" is non-standard terminology, possibly in analogy to the misnomer of "isentropic potential vorticity"? Please clarify.

pg 6, line 14ff: This is an important paragraph, as it introduces the role of elliptic Lagrangian structures. As is, however, it is unclear how this paragraph links to the rest of the discussion at this point in the manuscript. I recommend including a similar discussion in the introduction.

pg 7; LCS: This and several other acronyms are not defined. The concept of a LCS (Lagrangian coherent structure) is not introduced either.

pg 8; Lagrangian flow: unclear

pg 9: It would be very helpful to mark R1 and R2 in the figure. In general, I find the idea to follow circulation areas and their merging in a Lagrangian sense quite interesting. With the current presentation, however, the discussion does not provide much insight to the reader.

pg 9, line 15ff: I cannot follow the role of Lee described in this paragraph.

pg 9, line 29, 30; comment on Lagrangian conservation of vorticity. Why should vorticity be conserved materially?

pg 10, lobe transport of vorticity: It would be quite helpful for the reader to actually show figures including vorticity.

pg 14, line 8-9. Is the difference between 0.48 and 0.44 significant?

———————————————————

---

## Referee Comment (RC3) · Anonymous Referee #3 · 24 Mar 2017

In my opinion, this paper does not achieve its goal in illustrating how Nate interacts with its environment. Many results seem to be highly dependent on the way there are obtained and some statements are incorrect. It needs a major revision.

First, all the paper is based on the role of the air mass that comes from storm Lee. However, no figure is given to show the evolution and decay of this cyclone. In addition, there is no precise definition of the air constituting Lee, and it is then difficult to see which air mass will be involved in Nate development.

Second, the invariant manifolds may be highly sensitive on the way there are computed. From the different figures presented in the manuscript, small-scale motions may be very intense so as the exact position of the manifolds may change very much.

Now, my more precise comments.

[Figure]

1) Line 31, page 2.The definition of invariant manifolds based on a moving reference frame is wrong. Following your definition, any Galilean transform (e.g. rigid-body rotation) will change the position of saddle points and of the manifolds. The second definition given page 4 which relies on the Okubo-Weiss criterion is wrong as well, for the same reasons (see Lapeyre et al. Physics of Fluids 1999, Lapeyre et al. Chaos, 2001, Koh and Legras Chaos 2002, Haller JFM 2005). In the same manner, the authors cannot say page 14 that Okubo-Weiss is Galilean invariant! A correct definition of manifolds is given in the method section page 7.

2) It is quite difficult to follow the paper as one needs to understand the different air masses origin and and there is no synoptic view of Lee (add a figure, please!) and a definition of its air mass. Also, can the author show Nate in its embedded environment (i.e. in a much larger spatial region)? An example of my difficulty in reading the paper is given page 3, lines 13-14 when the authors state that "One or more vorticity filaments...". It would be very useful to see them! Same thing, about the S-shape (line 17). Can the authors illustrate the remnant air from Lee!

3) The discussion about the role of the Lee air mass in the genesis of Nate relies on the description of manifolds on isobar p=700hPa. However, Figs. 4, 5, 10 show that interpretations are highly sensitive to different parameters (altitude, divergence of the flow, SST...). Moreover, vertical motions are not included in the computation of the manifolds. It would be important to include these motions to see how the manifolds are dependent on this parameter as well. From what I see from the different figures, it is not clear to me that the positions in space of the manifolds are well defined. The very filamentary lobes may only exist because of advection by very small scales or errors in the velocity field. Manifold analysis is a powerful tool when the large-scale velocity fields is responsible of chaotic advection. Here, a lot of inertio-gravity waves seem to be emitted during convection and I wonder if they are quite energetic in terms of horizontal flow. If it is the case, that challenges a lot the interpretations of the paper.

4) Page 9, second paragraph. What is the true definition of air coming from Lee?
PV>some constant value ? air coming from latitude > 30 ? relative humidity < 50% ?

5) Panels in Figure 3 are unreadable. The color scale for PV does not highlight low and high values; also it is not possible to discriminate positive and negative values in OW criterion.

6) How are precisely defined R_1 and R_2? This is important to follow the interpretation.

7) The authors give average values of relative vorticity. However there are two subtleties. First, there is some uncertainty in the exact area of the lobes. This should be quantified. Second, there are a lot of gravity waves and I guess there are local spots with high values of vorticity. This can strongly affect the average value, so that the average would be meaningless.

8) Page 13. you should compare manifolds computed from trajectories along eta=0.6 and along p=600hPa surfaces to assess uncertainties in the position of the manifolds in the WRF simulation.

9) Figure 3 and 4 do not correspond to the same domain and the longitude axis is labelled differently. Please modify accordingly.

10) Page 13, line 18. "the flow on isobaric surfaces". I thought that it was on eta=0.6???

11) I don't see the point to paragraph about SST sensitivity. It does not seem to me that this paragraph is important for the discussion.

12) I do not agree with the discussion on the the vortex radial structure. First, how do you define an "average" radial profile? The vortex is not axisymetric at all. How is defined its center? From Figure 3, OW and PV are quite noisy due to convection, so radial average may be meaningless. I thus do not understand what is plotted in Fig.8.

Second, your definition of $u\sim$ and $v\sim$ is awkward. From the definition of T, we have $T(u\sim,v\sim) =(u,v)$ with $T=[u,n]/|u|$ Then $T(u\sim,v\sim)= (u\sim\ u\ -v\ v\sim\ ,\ u\sim\ v\ +\ u\ v\sim)$ Hence

$u\sim = |u|$ and $v\sim = 0$!

So I don't see why the use of $u\sim$ would be interesting.

13) What are the uncertainties on the curves in Fig 7.

14) Page 14,Line 21, the phrasing "vorticity moves inward" is misleading as it is not a 2D nondivergent transport. Also, it seems that the pouch boundary is defined through the OW criterion, which is quite different from the invariant manifold. Please clarify.

15) Panels of Fig. 10 should be at the same times as the Fig.2 Also, red/magenta colors are reversed with Fig.2

16) Conclusions. The fact that air cannot penetrate the vortex core while it can enter the pouch was discussed by Lapeyre Chaos 2002 and Babiano et al. Physics of Fluids 1994.

---

## Author Comment (AC1) · 27 Apr 2017

This is the author's responses to the review by Dave Ahijevych for "The genesis of Hurricane Nate and its interaction with a nearby environment of very dry air". We appreciate the reviewer's comments and have revised the manuscript so that all of these comments have been addressed. The reviewer's comments are given below in italic while the author's responses follow in regular font.

*General Comments: This paper extends the marsupial paradigm of tropical cyclogenesis to a non-African Easterly Wave. Without a wave-centric reference frame, air parcels around the storm are divided into two camps: those inside the pouch and outside. It is shown that some mixing occurs across the pouch boundary due to lobes breaking off, but dry air is shielded from the inner vortex by a shearing sheath that can be traced*

*back to the manifolds that separate pouch and non-pouch air. I almost cut my review short because I had never heard of the word 'manifold' outside of a car engine setting, but after reading through the paper, which is fairly well written, I now have a decent idea of what it means. Now that I've read the paper, the abstract and introduction make much more sense. Later in the paper there is a weaker section with some mistakes in the figure and text, but overall this is ready for publication. My technical corrections are the main component of this review. Some of the paper is still way beyond my area of expertise, and I have no idea whether it is right or wrong, such as the section on 'Manifold computations' (pages 7-8).*

While the use of Lagrangian manifolds and lobe transport is likely not familiar to many readers, these concepts are important for understanding transport on stratified isosurfaces in a time-dependent flow. The revised version provides a more thorough introduction so that a reader without expertise in Lagrangian manifolds can follow more easily. Many other improvements have also been made including making the results section much more readable and better demonstrating the role of Tropical Storm Lee in providing vorticity to the pre-Nate region.

*Specific comments:*

*After rereading the abstract it isn't clear why source the 'tilting' mechanism is given such prominence. It is barely mentioned again in the paper.*

We have removed mention of the tilting mechanism from the abstract.

*The adjective 'layer-wise' used in the abstract and other places is confusing. Could you replace 'layer-wise advection' with 'advection on a constant pressure level'?*

We have replaced 'layer-wise' with 'advection on a constant pressure level' and removed the term 'layer-wise' from the manuscript.

*Page 3, line 5. It isn't clear what 'non-advective fluxes' are. This becomes apparent later on, but it is first mentioned here without any explanation.*

We have explained what non-advective fluxes are at this point in the manuscript.

*Page 9, line 15. It is not obvious that the pouch region that originated from Lee still contains high potential vorticity air. From the figure, PV is muddled and similar to the PV in the portion with Gulf origins.*

A new figure has been added that shows the PV from Lee from an earlier time, and the attracting line that differentiates the regions $R_1$ and $R_2$. The air contained in $R_1$ clearly shows the source high PV air from Lee that enters the pre-Nate region.

*Technical corrections:*

*Page 3, footnote 2. Change 'if' to 'of'.*

We have made the correction.

*Page 7, Figure 1 caption. Change 'L2' to 'L4'.*

We have corrected the labels in the figure and text.

*Page 7, line 9. What is 'LCS'?*

LCS is an acronym for Lagrangian Coherent Structure, which we have added to the paper.

*Page 7, line 10. What is 'FTLE'?*

FTLE is an acronym for Finite-time Lyapunov exponent, which we have added to the paper.

*Page 8, line 29. U2 is never labeled in the figure. That would be nice.*

U2 is now labeled in the figure.

*Page 9, line 2, 3rd-to-last word. Change 'northeast' to 'southwest'.*

We have made the correction.

*Page 9, line 6, last term should be S1(I4,H1), not S2(I4,H1).*

We have made the correction.

*Page 9, footnote 7. What is 'VS'?*

VS is an acronym for vorticity substance, which we have added to the paper.

*Page 10, line 19. Can the lobe labels of L3 and L4 in Figure 2 (a) be reversed so as to match the order of the labels in Fig. 1? In other words, change L3 to L4 and change L4 to L3?*

We have reversed the lobe labels on Figure 1 to match those in the remainder of the paper.

*Page 10, line 25. Explicitly reference Fig. 2 after mentioning the gradient of theta-e. I was incorrectly looking for it (theta-e) in Fig. 3.*

A reference to Figure 2 has been added where $\theta_e$ is mentioned.

*Page 10, line 26. 'north' should not be capitalized.*

We have made the correction.

*Page 12, Figure 3 caption. Is the satellite panel described accurately? The text says it is 0.6 um, but isn't that visible? Wouldn't that be white for clouds? Not sure what the units are of the color table and it isn't clear how it could be 0.6 um imagery.*

The GOES imagery shown is 3.9 $\mu m$ shortwave radiation, and cold clouds are indicated by lower values. We have converted the data to temperature (K) in the revised paper. The units and description have been corrected in the text and the caption.

*Page 13, Line 21. Manifolds are overlaid on water vapor not vorticity in Figure 5.*

We have made this correction to indicate that the manifolds are overlaid on water vapor.

*Page 14, first paragraph. Fig doesn't seem to go with text. Text talks about 700 mb*

*vorticity and area. Figure has circulation and mean vorticity. Figure would also benefit from un-squished text. Y-axis ranges of panel (a) should be same as panel (b).*

The Figure reference in the text has been changed to Figure 10. Panels (a) and (b) have been combined onto a single plot so that a comparison between the two simulations can be more easily made.

*Page 14, line 18. Panel labels don't start with (a) in text reference or actual Figure 8.*

We have corrected the figure labels.

*Page 14, last paragraph. This paragraph is confusing. In line 21, when you say 'as the vorticity moves inward', do you mean the vorticity maximum (in the radial profile) is moving inward? And in line 27 what is meant by 'the stability of the unstable manifold'?*

We have changed the text to indicate that vorticity increases toward center due to the convergent flow. We were referring to the stability of the flow along the unstable manifold, which switches from stretching to shearing during entrainment. This point has been clarified in the revised manuscript.

*Fig 10, and Page 15, line 28. I think this time range is Sep 6 to 8, but can't read the time labels on some panels.*

We have corrected the time range in the text and in the figure caption, and have moved the time labels or changed colors in many of the figures so that they are more readable.

*Page 16, line 6. The term 'limit cycle' is first used. I was not familiar with it, but I understand it now after reading further.*

The term limit cycle is now clearly defined at this point in the paper.

*Page 16, lines 8-10. Something with wrong with references to Fig. 9 (c)-(h), which don't exist.*

The reference should be to Figure 10 (c)-(h) and this error has been corrected.

*Page 20, Fig. 8. Need units in figure or caption.*

We have added units in the caption.

---

## Author Comment (AC2) · 28 Apr 2017

This is the author's responses to the review by Anonymous Referee #2 for "The genesis of Hurricane Nate and its interaction with a nearby environment of very dry air". The authors would like to thank the referee for comments, and we feel that we have been able to address all of the areas of concern in the revised version of this paper. These comments have been used to improve the quality of this paper, and our responses to each of the individual comments is given below. The referee comments are given in italic while the author's responses are given in standard font.

*Validity of 2D analysis: The manuscript provides little motivation why analysis of La-grangian boundaries of the quasi-horizontal flow should provide insight into the devel-opment of Nate (2011). Certainly, the authors make some assumptions that they had*

*articulated in earlier work. These assumptions, however, should be stated also in this manuscript. Importantly, the 2D assumption seems to be in contrast with a statement made by the authors about the role of convection in lobe transport (pg 13, line 28). Clarification is required. Nate develops near the boundary between two air masses. The authors emphasize the large moisture gradient across this boundary. Arguably, the very dry air to the North of Nate is of midlatitude origin and presumably this air mass is also considerably colder than the tropical air mass in which Nate develops. In short, I expect a strong baroclinic zone to the North of Nate and the large-scale, 2D flow follows isentropic rather than isobaric surfaces. I question that the analysis on isobaric surfaces is indeed Lagrangian, in the sense that the authors follow air parcels transported by the large-scale (adiabatic) flow, which is most likely one of the nonarticulated assumptions made by the authors. Convincing justification for the use of an isobaric framework in the presence of strong baroclinicity is needed.*

We agree with the reviewer that both baroclinic contributions and vertical motions from convection bring into question the validity of the assumption of 2D velocities. Convection acts to increase the vertical vorticity while decreasing the area and leaving the circulation within any Lagrangian loop, including lobes, unchanged. Thus, the use of 2D velocities in a flow with convection still captures the advective changes to the circulation. The use of isobaric coordinates rather than isentropic coordinates does not change the validity of horizontal velocities used to advect material curves, but only contributes a non-advective flux of vorticity proportional to the pressure vertical velocity (Haynes and McIntyre, 1987). Along the unstable manifold segment that is aligned with the frontal boundary, the tilting flux is approximately 20% of the magnitude of total vorticity flux which includes the advective flux and the increase in circulation due to contraction of the vortex. The precise computation of these fluxes will be discussed in greater detail in a future paper. In the revised manuscript, we have included isentropic manifolds at the 315 K level, and notice that despite some vertical motions along the frontal boundary, there is no topological change in the manifolds, and only small changes in the fine structure.

*The authors focus on the objective identification of hyperbolic Lagrangian structures. Similar methods can be applied to identify elliptical Lagrangian structures, which play an important role in separating a vortex from its environment also. The authors appreciate this role by their qualitative discussion of the 'shear sheet', e.g. on page 6. For a wave's critical layer (e.g. Dunkerton et al. 2009) there is conceptual understanding why the flow boundaries that arise from the environmental flow, and are thus the relevant boundaries, in which the embryo tropical cyclone develops, are hyperbolic structures. Such conceptual background misses for non-AEWs disturbances like pre-Nate, or at least the authors do not provide such background. Therefore, an objective identification of elliptical boundaries will considerably strengthen this manuscript. In addition, the identification of elliptic boundaries would help to introduce the concept of a limit cycle, which is referred to later in the manuscript, and help to define the core of the disturbance, which is undefined in the current version of the manuscript.*

We thank the referee for suggesting a further discussion of elliptic Lagrangian boundaries. Much of our discussion in this paper is on the hyperbolic boundaries that are present when there is no distinguished reference frame or hyperbolic structures in that reference frame resulting from the wave flow. The elliptic boundaries are present in all cases of cyclogenesis whether a parent wave is present or not, and in the case of AEW flows, the elliptic structures are located close to closed streamlines in the wave-relative frame interior to the hyperbolic structures. In mature cyclones, elliptic boundaries do play a role in protecting the vortex from its environment because hyperbolic structures do not persist until the point of axisymmetrization. However, elliptic structures protect a developing vortex core from air that has passed through the outer pouch boundary. We have added a description of objective elliptic boundaries, their mathematical definitions, and the concept of a limit cycle to Section 2. These elliptic boundaries are now shown along with the manifolds in Figure 2. The location of the regions of high shear and of solid-body rotation at the core support our previous discussion, and help to show how the shear sheath interior to the outer pouch and external to the core protects the core from air that has penetrated the outer pouch.

*pg 1, line 15; vorticity generation by tilting: This aspect is hardly touched on in this manuscript. I recommend omitting reference to this process in the abstract (and in the conclusions).*

We have removed reference to the tilting mechanism in the abstract, and 'tilting' does not appear in the conclusions.

*pg 5, line 1; and elsewhere; 'Eulerian boundary': There are references in the manuscript to Eulerian streamline patterns that are not illustrated in this manuscript. In addition, there are references to the role of tropical cyclone Lee that are not illustrated either. For the reader, it is rather hard to follow (and appreciate) these descriptions. I suggest using one or two additional figures to illustrate such points; or to keep such references to a minimum.*

We have added an additional figure (2a) showing TS Lee and the regions of high PV from Lee that contribute to Nate. The Eulerian streamlines are not important in any part of our analysis, and there is no distinguished reference frame in which to view the Eulerian streamlines. so we have not included them. The Lagrangian boundaries from Lee make it clear that the Lagrangian boundaries, and not the Eulerian boundaries, are those relevant for the transport of vorticity from the Lee flow to the Nate development region.

*pg 7, The subsection 'Manifold computations' requires considerable improvements: The authors use phrases like 'some situations' and 'additional options' but it remains unclear if or when other options are used or what methods are applied in other situations. Most importantly, it remains unclear from this description for how long the underlying trajectories have been calculated. It is well known that finite-time Lagrangian coherent structures are sensitive to the integration time. A more explicit discussion of this integration time and a discussion of the sensitivity of the results to integration time are needed.*

This section has been improved by removing vague statements and by explicitly stating

the length of integration time used in both in obtaining the initial curve segments and in advecting the curves. We now also discuss the sensitivity of Lagrangian boundaries to integration time and explain that a 2-3 day integration is required to achieve a closed pouch, while longer integrations increase the amount of filamentation.

*pg 10, 'Relation of Lagrangian ...': Unfortunately, the presentation of the results deteriorates rather significantly from here on. E.g., the authors note that PV and O3 is shown and then continue with a discussion of $\theta_e$, the GOES imagery is presented without units, convergence is presumably confused with confluence (pg. 13, line 1), results from WRF at 600 mb (which should read hPa) are compared to results from ECMWF data at 700 hPa, vorticity is confused with mixing ratio in Fig. 5, it is unclear what the difference is between individual panels in Figs. 4-6, : : : The subsection 'Backward trajectories' is very dense and it seems as some important information is not given to the reader. I cannot identify in the figures several features described by the authors. This is of particular importance with respect to the vertical similarity of manifolds and the limit cycle.*

The results from page 10 on have been improved significantly. In particular, the above concerns have all been addressed in the revised manuscript. PV and O3 are now discussed where they are first mentioned, and the discussion of $\theta_e$ now follows after that discussion. We have converted the GOES data to temperature (K), and corrected the description and units in both the text and the figure caption. We have changed 'convergent' to 'confluent' to describe the flow along the unstable manifold.

We have now computed the manifolds for the WRF simulation at the $\eta = 0.7$ level to make a proper comparison with the ECMWF 700 hPa analysis, and the analysis at $\eta = 0.7$ has also been included for the SST sensitivity and non-divergent experiments.

The text reference to Figure 5 has been corrected to state that the mixing ratio is shown. The different panels in Figures 4 and 6 now show time-evolving manifolds overlaid on vertical vorticity. The labels on the figures and reference in the text have been improved

in the new manuscript. The time labels are now easier to read so that the individual panels are easier to distinguish.

The backward trajectory section has been expanded to make it easier to follow, and each of the panels is now discussed individually and in greater detail. We have also provided further details on how the trajectories were integrated. Key features have been labeled on the trajectory plots.

*The comparison between the ECMWF and the WRF data is confusing. Importantly, it is not clear how much the results based on the ECMWF data can be trusted. Furthermore, the comparison of the results using the full wind field and the non-divergent flow only is poorly motivated.*

The non-divergent wind field is used to demonstrate that convergent flow is not necessary for a pouch boundary to be closed, and is unnecessary for the formation of lobes. However, the size of the lobes that intrude is far greater for convergent flow. We have made the comparison between the ECMWF data and the WRF data more clear, and now state that the ECMWF data show large non-advective fluxes, but has approximately the same topology as the WRF data. While the resolution of the ECMWF data may lead to errors in the fine-scale filaments, the manifolds still closely follow the gradients of tracers such as $O_3$, indicating that this data is sufficient to capture the larger-scale transport including the lobe transport prior to extreme filamentation.

*The conclusions refer to several aspects that have not been discussed sufficiently in the manuscript. The arguments given in enumerations 1) and 2) are plausible but have not been shown in this manuscript. The 'core' referred to in enumeration 5) has never been defined. Finally, the Eulerian streamlines noted on pg 19 have not been shown in this manuscript. The revised conclusions should focus much more on results and insight that is shown and developed in the manuscript at hand.*

With the improvements in the revised manuscript, all of the ideas discussed in the Conclusions have now been discussed in the main manuscript in sufficient detail. Arguments 1) and 2) in the enumeration are now shown more clearly in the manuscript and supported by the addition of the Figure 2a showing the attracting line coming from the Lee flow dividing $R_1$ and $R_2$. The inclusion of elliptic structures demonstrates core formation, and the core is defined earlier in the manuscript at the end of Section 2.

*Technical corrections/ Editorial recommendations: pg 1, line 2: of > or*

We have made the correction.

*pg 2, line 33; kinematic structures as consequence of invertibility of vorticity: I cannot follow this argument. Please clarify.*

We have clarified the statement to indicate that non-AEW disturbances provide no distinguished frame of reference.

*pg 3, line 2; 'arm of T.S. Lee': incomprehensible*

We have replaced 'arm' with the 'curved vorticity filaments emanating from' Lee.

*pg 3, line 5: should read 'for a non-AEW disturbance'.*

We have made the correction.

*pg 3, line 12; suggest: initiated > first identified*

We have made the suggested correction.

*pg 5: should 'vortex strip' read 'vorticity strip'?*

We have made the suggested correction.

*pg 5, line 20: according to the references it should read 'Rutherford and Dunkerton, 2017.'*

We have corrected the reference.

*pg 6, line 4: This sentence seems to lack something, maybe 'vorticity' after 'system'?*

We have changed the wording to 'system circulation' .

*pg 6, line 5; 'isobaric vorticity substance' is non-standard terminology, possibly in analogy to the misnomer of 'isentropic potential vorticity'? Please clarify.*

We have changed the wording to 'isobaric absolute vorticity'.

*pg 6, line 14ff: This is an important paragraph, as it introduces the role of elliptic Lagrangian structures. As is, however, it is unclear how this paragraph links to the rest of the discussion at this point in the manuscript. I recommend including a similar discussion in the introduction.*

We have added a definition of elliptic structures earlier in the paper which includes the Lagrangian vorticity.

*pg 7; LCS: This and several other acronyms are not defined. The concept of a LCS (Lagrangian coherent structure) is not introduced either.*

We have added the meaning of the acronym LCS and an explanation of what an LCS is. We have also clearly defined all other acronyms used in the paper.

*pg 8; Lagrangian flow: unclear*

We have changed 'Lagrangian flow' to 'Lagrangian manifolds'.

*pg 9: It would be very helpful to mark R1 and R2 in the figure. In general, I find the idea to follow circulation areas and their merging in a Lagrangian sense quite interesting. With the current presentation, however, the discussion does not provide much insight to the reader.*

The new Figure 2a showing the potential vorticity from Lee also shows the regions $R_1$ and $R_2$ and the curve that separates them, so that it is clear what regions the circulation values refer to.

*pg 9, line 15ff: I cannot follow the role of Lee described in this paragraph.*

The revised manuscript includes a new Figure 2a demonstrating the regions of high potential vorticity that originated from Lee, and this paragraph has been edited to point the reader to the key features in the figure.

*pg 9, line 29, 30; comment on Lagrangian conservation of vorticity. Why should vorticity be conserved materially?*

We have changed the wording so that this sentence could not be interpreted to mean that models other than the ECMWF model conserve vorticity.

*pg 10, lobe transport of vorticity: It would be quite helpful for the reader to actually show figures including vorticity.*

A figure showing the potential vorticity from Lee has been added in Figure 2a, and Figures 4 and 6 now show vorticity from the WRF simulations.

*pg 14, line 8-9. Is the difference between 0.48 and 0.44 significant?*

We have changed this section to reflect the analysis on the $\eta = 0.7$ level. The values of area reduction have been changed to reflect the new results, and the new text indicates that the difference between the two simulations is small.

---

## Author Comment (AC3) · 1 May 2017

The authors would like to thank the reviewer for the detailed comments that we have used to improve the quality of this paper. We have written a new version of the manuscript that takes all of these comments into consideration. The responses to each of the individual comments is given below. The reviewer's comments are given below in italic while the author's responses are given in standard font.

*In my opinion, this paper does not achieve its goal in illustrating how Nate interacts with its environment. Many results seem to be highly dependent on the way there are obtained and some statements are incorrect. It needs a major revision. First, all the paper is based on the role of the air mass that comes from storm Lee. However, no figure is given to show the evolution and decay of this cyclone. In addition, there is no*

*precise definition of the air constituting Lee, and it is then difficult to see which air mass will be involved in Nate development. Second, the invariant manifolds may be highly sensitive on the way there are computed. From the different figures presented in the manuscript, small-scale motions may be very intense so as the exact position of the manifolds may change very much. Now, my more precise comments.*

We thank the reviewer for the detailed comments and we have revised the manuscript that considers all of the reviewer's comments. We have made substantial revisions to the paper that strengthen the results and have corrected the errors. In particular, we have added Figure 2a showing the potential vorticity from Lee and the Lagrangian boundaries associated with the transport of this high potential vorticity air into the Nate development region, a precise definition of the region containing high potential vorticity from Lee, and the subsequent formation of the boundary associated with the sharp moisture gradient to the north of the Nate vortex. While the manifolds are sensitive to the way they are computed, e.g. due to the choice of the time interval used for integration, the sensitivity affects filaments that have little circulation, but does not affect the larger scale transport, so our major conclusions hold regardless of small scale uncertainties. We have added more precise statements about how the manifolds are computed, and have added an additional discussion about the sensitivity in Section 4.

*1) Line 31, page 2.The definition of invariant manifolds based on a moving reference frame is wrong. Following your definition, any Galilean transform (e.g. rigid-body rotation) will change the position of saddle points and of the manifolds. The second definition given page 4 which relies on the Okubo-Weiss criterion is wrong as well, for the same reasons (see Lapeyre et al. Physics of Fluids 1999, Lapeyre et al. Chaos, 2001, Koh and Legras Chaos 2002, Haller JFM 2005). In the same manner, the authors cannot say page 14 that Okubo-Weiss is Galilean invariant! A correct definition of manifolds is given in the method section page 7.*

We find this comment to be a misinterpretation of our definition on page 2, as the Lagrangian manifolds used in the analysis of Nate are not defined to be those of a

stagnation point in the moving frame as clearly, the manifolds only coincide with Eulerian streamlines if one can assume that the flow and reference frame are steady. We have modified the text on page 4 to make it clear that a distinguished frame is still assumed at this point. The assumption of a steady flow in a moving frame of reference used in the definitions on page 2 and 4 are not used in any of the results in this study, and only serve as a motivation for the use of the correct definition on page 7. We have changed the wording on pages 2 and 4 to indicate that these manifold definitions are only valid in the case of steady flows where the frame is specified. The wording on page 14 for the Okubo-Weiss criterion has been changed to 'translation invariant'.

*2) It is quite difficult to follow the paper as one needs to understand the different air masses origin and and there is no synoptic view of Lee (add a figure, please!) and a definition of its air mass. Also, can the author show Nate in its embedded environment (i.e. in a much larger spatial region)? An example of my difficulty in reading the paper is given page 3, lines 13-14 when the authors state that "One or more vorticity filaments...". It would be very useful to see them! Same thing, about the S-shape (line 17). Can the authors illustrate the remnant air from Lee!*

The new Figure 2a showing Lee and the pre-Nate region demonstrates the interaction of these different air masses. The vorticity filaments and s-curve connecting the Nate and Lee flows that are described in the paper are now shown in the figure.

*3) The discussion about the role of the Lee air mass in the genesis of Nate relies on the description of manifolds on isobar p=700hPa. However, Figs. 4, 5, 10 show that interpretations are highly sensitive to different parameters (altitude, divergence of the flow, SST...). Moreover, vertical motions are not included in the computation of the manifolds. It would be important to include these motions to see how the manifolds are dependent on this parameter as well. From what I see from the different figures, it is not clear to me that the positions in space of the manifolds are well defined. The very filamentary lobes may only exist because of advection by very small scales or errors in the velocity field. Manifold analysis is a powerful tool when the large-scale*

*velocity fields is responsible of chaotic advection. Here, a lot of inertio-gravity waves seem to be emitted during convection and I wonder if they are quite energetic in terms of horizontal flow. If it is the case, that challenges a lot the interpretations of the paper.*

We agree that the manifold filaments are sensitive to integration time, vertical level, divergence, SSTs, moist processes, or inertio-gravity waves. The contents of these filaments may also be quickly diffused, particularly when entrained into the core. While we show these filaments, the discussion in this paper is concentrated on the entrainment of the lobes from the dry air region which is not sensitive, as it appears in both ECMWF and WRF, with constant and varying SSTs, and aligns very closely with isolines of $O_3$. Thus, the key structures that we highlight are driven primarily by the large-scale velocity field, and only the filamentary regions, which have very little circulation, are strongly influenced by the small scales. A comparison of Figure 4 and 6 shows very similar manifolds even under separate model simulations. We have further explored the role of vertical motions by computing the manifolds in isentropic coordinates, and the same conclusions hold, even though the filamentary details are again different. The comparison of the non-divergent flow in Figure 5 to Figures 4 and 6 should not be seen as a sensitivity analysis but rather the demonstration of a dynamic process, since the non-divergent manifolds were created using only the non-divergent part of the velocity field, and not by turning off divergence in the WRF model. To clarify this issue, the new manuscript includes further discussion on the verification of robustness of large-scale Lagrangian structures. Specifically, we have added a paragraph of text when introducing $O_3$ and also in the Conclusions.

*4) Page 9, second paragraph. What is the true definition of air coming from Lee? PV>some constant value ? air coming from latitude > 30 ? relative humidity < 50% ?*

The air coming from Lee is that coming from above the attracting LCS within the pouch that is shown in the new Figure 2 (a) in the region labeled $R_1$.

*5) Panels in Figure 3 are unreadable. The color scale for PV does not highlight low and*

*high values; also it is not possible to discriminate positive and negative values in OW criterion.*

We have changed the text color where needed to make the figures more readable, and have changed the color scale on the PV and OW plots to highlight low and high values.

*6) How are precisely defined $R_1$ and $R_2$? This is important to follow the interpretation.*

$R_1$ and $R_2$ are precisely defined by closed Lagrangian loops comprised of specific manifold segments terminating at specific endpoints and the regions are separated by an attracting boundary between them shown as a yellow curve in the new Figure 2 (a). The circulation values of these regions are precise because of the exact locations of the Lagrangian curves bounding them.

*7) The authors give average values of relative vorticity. However there are two subtleties. First, there is some uncertainty in the exact area of the lobes. This should be quantified. Second, there are a lot of gravity waves and I guess there are local spots with high values of vorticity. This can strongly affect the average value, so that the average would be meaningless.*

While there may be uncertainty in the manifold locations due to the spatio-temporal resolution of velocities, there is almost no uncertainty in the area of lobes or of the circulation since both the area and circulation are computed by a contour integral along the manifold using Green's Theorem and Stokes' Theorem, respectively. Mean vorticity is then computed as the ratio of circulation and area. Of course, the vorticity distribution is not constant, but the mean vorticity or circulation is a common and meaningful diagnostic of tropical cyclone strength.

*8) Page 13. you should compare manifolds computed from trajectories along eta=0.6 and along p=600hPa surfaces to assess uncertainties in the position of the manifolds in the WRF simulation.*

We now show the ECMWF at 700 hPa and WRF simulation at $\eta = .7$ for our primary

analysis so that a more meaningful comparison between the ECMWF and WRF data can be made.

*9) Figure 3 and 4 do not correspond to the same domain and the longitude axis is labelled differently. Please modify accordingly.*

We have modified the figures so that the WRF and ECMWF plots show the same domain.

*10) Page 13, line 18. "the flow on isobaric surfaces". I thought that it was on eta=0.6???*

We have changed the text to indicate that the flow is on the eta=.7 level, where the new analysis is performed to make the WRF and ECMWF analysis more consistent.

*11) I don't see the point to the paragraph about SST sensitivity. It does not seem to me that this paragraph is important for the discussion.*

The forecast discussions for Nate indicate that actual intensity was lower than forecast intensity. Environmental interaction and lower SSTs from upwelling were given as possible causes for the lower intensity, and the paragraph about SST sensitivity shows the impact of lower SSTs in the WRF model. We demonstrate that the SST difference has little impact on the manifold configurations and only a modest impact on the strength of the vortex. The SST sensitivity paragraph has been revised to make the importance of this simulation more apparent.

*12) I do not agree with the discussion on the the vortex radial structure. First, how do you define an "average" radial profile? The vortex is not axisymetric at all. How is defined its center? From Figure 3, OW and PV are quite noisy due to convection, so radial average may be meaningless. I thus do not understand what is plotted in Fig.8.* Second, your definition of $u$ and $v$ is awkward. From the definition of $T$, we have $T(u,v) = (u,v)$ with $T = (u,n)/|u|$ Then $T(u,v) = (uu, -vvv, uv + uv)$ Hence $u = |u|$ and $v = 0$ So I don't see why the use of $u$ would be interesting.

While the vortex is not axisymmetric, the flow near center is nearly circular so that radial

profiles for small radii are meaningful. The center location is unambiguous as there is almost no difference between the locations from the best-track data set and the MRG pouch products, which were computed independently. The transformation matrix T orients the velocity field to give the tangent and normal components, thus $\tilde{u} = |\mathbf{u}|$ and $\tilde{v} = 0$. However, T is used to project derivatives of the velocities onto the tangent and normal directions, providing a 'natural coordinate' definition of normal strain and shear strain. We have made this more clear in the revised manuscript.

*13) What are the uncertainties on the curves in Fig 7.*

The curves in Figure 7 show the circulation interior to the manifolds, and the circulation is taken as a line integral along the curve, which has point spacing <=.01 degrees. There is no visible variation in the curves using a larger point tolerance. Any uncertainty in the circulation would be based on the location of the manifolds, and this sensitivity is discussed further in the conclusions, as indicated by our response to a previous concern.

*14) Page 14,Line 21, the phrasing "vorticity moves inward" is misleading as it is not a 2D nondivergent transport. Also, it seems that the pouch boundary is defined through the OW criterion, which is quite different from the invariant manifold. Please clarify.*

The flow near the center is not 2D non-divergent transport as upward mass flux causes convergence toward center, and we have changed the wording of this statement to indicate that vorticity increases toward the center due to convergence. The pouch boundary is defined through the invariant manifold locations, and not by the OW criterion, and the wording has been changed to make this point more clear. High OW is a characteristic of the core, and negative OW is characteristic of the shear sheath, not of the pouch boundary.

*15) Panels of Fig. 10 should be at the same times as the Fig.2 Also, red/magenta colors are reversed with Fig.2*

We have changed the colors on Figures 1 and 10 and times on Figure 10 to match those in Figure 2 so that the manifold colors are the same on all figures.

*16) Conclusions. The fact that air cannot penetrate the vortex core while it can enter the pouch was discussed by Lapeyre Chaos 2002 and Babiano et al. Physics of Fluids 1994.*

We have added a discussion including the results of these studies and references at the point in the paper where the vortex core is first defined in Section 2.

---

## Referee Report (RR1)

The authors have done a great effort to substantially improve their manuscript. Their results are now better explained with a much more precise wording.

I still request some important corrections before publication.

Major comments:

1. I appreciate the effort of the authors to display the separation of air masses $R_1$ and $R_2$ in Fig.2a and the manifold related to Lee. However I still find the presentation of the results rather confusing. In particular, I have a lot of difficulties in understanding the precise (not qualitative) definitions of the pouch, of the different regions R1, R2, relation with the lobe L1 or the air coming from Nate).

    (a) It would be useful to have a figure at an earlier time (for instance 12 UTC Sept. 5 since it is mentioned in the text line 19, page 10) to see the appearance of L1.

    (b) How are defined $R_1$ and $R_2$ at 12UTC 7Sept? Is it just $L_1$ and the pouch, respectively? Do you track the manifold related with Lee? The separation curve should be displayed on Fig. 2d or 2e. This is quite important in your discussion page 10.

2. Line 15, page 10, it is claimed that the pouch region originating from Lee contains high PV air while line 23 page 11, it is claimed that L1 contains low PV air.

    But

    (a) I am completely unable to deduce these two statements from Fig. 3b. Instead of having a continuous gray shading, can you map only a limited number of gray nuances? Also can you show only positive PV values?

    (b) Also isn't it contradictory that L1 and R1 have air masses with different properties if both represent the lobe region? I guess this is the main problem of the manuscript as it is not precise enough to distinguish R1 and the pouch region, so that in many places the reader is confused between the two regions.

3. Concerning the non-divergent lobe transport, line 31, page 15, you cannot claim that "divergence has little impact on manifold structure" since, precisely, the non-divergent alone cannot explain the size of the lobe!

    Actually, it is quite interesting to see that the lobes almost completely disappear (contrary to what you state) and are parallel to contours of the water vapor mixing ratio. This should be discussed as it means that the velocity divergence is responsible of the convergence of the material lines towards the center of the cyclone and of the inward transport of dry air. On the contrary, non-divergent motions seem related to a transport barrier so that no air from outside can enter inside the vortex.

4. Last statement page 16 is incorrect: I am unable to see a quick drop in relative humidity outside the shear sheath. Values are still quite high (r> 85%) at radius r= 2.5.

5. Your Okubo-Weiss definition does not take into account the divergent part of the flow. It has to be taken into account in addition to the shearing effects since it brings material inwards, even if there is a "shear sheath". This is quite important to discuss, as even hyperbolicity is ill-defined in this context.

Minor comments:

1. Top of page 10. In ECMWF, L1 has a negative circulation. On the contrary, when looking at Fig. 4a, L1 in WRF simulation is positive. Please comment.

2. The color palette is also awful for (relative?) vorticity in Fig.4. Can you use something that would be more practical? Same thing for Fig.3. All panels are kind of "blurred". Also in Fig.4, the color of the manifolds is reversed compared to Fig.2.

3. Page 16, line 17. It is incorrect to say that "the strain regions are advected inwards". There is no advection in here!

4. You still do not explain in the text how results of figure 7 are obtained (radial average around the center defined by ...).

5. Figure 7. You should say that the radius is in degree.

---

## Author Response (AR2)

**Author's responses to review by Anonymous Referee #3 for "The genesis of Hurricane Nate and its interaction with a nearby environment of very dry air"**

Blake Rutherford [1], Timothy Dunkerton [1], Michael Montgomery [2], and Scott Braun [3]

[1]Northwest Research Associates, Redmond, WA
[2]Naval Postgraduate School, Monterey, California
[3]NASA

*Correspondence to:* Blake Rutherford (blake@nwra.com)

We have considered all of the comments made by both reviewers, and have used those comments to produce a revised manuscript. In the revised manuscript, the largest difference from the previous is a new Figure 2 that shows the location of the attracting line separating the two regions within the Nate pouch on Sept. 5. This line is then tracked through the remainder of the time interval, and it is shown again at a later time in Figure 3. This figure helps to address the concern of one of the reviewers that the regions comprising the Nate pouch were not clearly marked, and the confusion that the region that we call $R_1$ was in fact the same as the region that we call $L_1$. New labels added to Figure 3 (a) make it very clear that these are separate regions.

In the revised manuscript, added text is given in blue, while deleted text is shown in red. The reviewer's comments are listed below while our responses follow in blue.

Reviewer 1:

Minor comments:

* end Section 2: KAM torus needs introduction/ references

We have added references for the KAM torus and its finite-time Lagrangian version.

* end Section 2: The definition of vortex core (Eq. 4) is unclear. Is there a threshold to be exceeded for the Lagrangian vorticity to qualify as vortex core? In addition, it should be noted that this is only one of several possible definitions of a vortex core.

We have changed the text to indicate that we have chosen to use a finite-time Lagrangian definition of a vortex core, and have stated that the vortex core must have positive cyclonic Lagrangian averaged vorticity. This definition does not require any certain thresholds to be met, only that the vorticity be positive.

* Section 4, 2nd line: Unclear how southerly flow can advect feature of Lee when that system is to the northeast. Please clarify.

We have corrected this statement to indicate that northerly flow was responsible for the vorticity advection.

\* Section 4, 2nd paragraph: The authors have agreed in their response to the first review that discussion of the Eulerian frame is not important here. I thus suggest deleting this paragraph, as it is hardly helpful to the reader without an associated figure anyways.

We agree with the suggestion and have eliminated this paragraph.

\* caption Fig. 2, first line: M ? K

We have made the correction.

\* last sentence in Section 4g: I disagree with the authors statement. To me, a distinct gradient in relative humidity associated with the shear sheath is not evident in Fig. 7c. I suggest deleting the sentence.

We have deleted this statement, as this suggestion was also made by the other reviewer.

\* conclusions, third paragraph; When Nate was a tropical storm in close proximity to dry air, the entrainment of dry air was limited to what was transported by lobes when an enclosed Lagrangian boundary was present: The relative role of lobe transport and the direct pathway at 850 hPa is discussed more completely in the paragraph after the enumeration. The simple statement here ignores the role of the direct pathway. I suggest deleting the statement here, as it is incomplete and there is redundancy with the later discussion anyways.

We agree with this suggestion and have eliminated this paragraph.

\* conclusions, enumeration 1) and 2) suggest an interesting link between the hyperbolic manifolds and the underlying dynamics. Can a similar link be provided for the existence of lobes?

The dynamical connection to lobes occurs when folding of vorticity filaments near a saddle causes the alternating exchange of material into and out of the cat's eye. This point is now highlighted in point 3 of the enumeration.

\* conclusions, first paragraph after the enumeration: I suggest deleting the reference to the Eulerian streamlines, consistent with elimination of references to an Eulerian frame throughout the revised manuscript.

We agree and have eliminated the comparison between Lagrangian manifolds and Eulerian streamlines.

Reviewer 2:

Review of the revised manuscript by Rutherford et al. The authors have done a great effort to substantially improve their manuscript. Their results are now better explained with a much more precise wording. I still request some important corrections before publication.

We appreciate the reviewer's comments, and have revised the manuscript to address these remaining concerns. Specifically, we have taken care to make the definitions of $R_1$ and $L_1$ much more clear by adding a new Figure.

Major comments: 1. I appreciate the effort of the authors to display the separation of air masses R1 and R2 in Fig.2a and the manifold related to Lee. However I still find the presentation of the results rather confusing. In particular, I have a lot of difficulties in understanding the precise (not qualitative) definitions of the pouch, of the different regions R1, R2, relation with the lobe L1 or the air coming from Nate). (a) It would be useful to have a figure at an earlier time (for instance 12 UTC Sept. 5 since it is mentioned in the text line 19, page 10) to see the appearance of L1. (b) How are defined R1 and R2 at 12UTC 7Sept? Is it just L1 and the pouch, respectively? Do you track the manifold related with Lee? The separation curve should be displayed on Fig. 2d or 2e. This is quite important in your discussion page 10.

The regions R1 and L1 are separate regions, as R1 is in the interior of the pouch at 0 UTC 6 Sept. and L1 is still outside. An
35    additional label of L1 in Figure 3 (a) and a comment where L1 is first introduced have been added to clarify this issue.

(a) An additional figure showing the attracting line at an earlier time has also been added.

(b) With R1 clearly distinguished from L1, it is clear that the discussion on page 10 is about the 2 separate regions within
the pouch. The location of the curve separating these regions has been added to Figure 2 (d).

2. Line 15, page 10, it is claimed that the pouch region originating from Lee contains high PV air while line 23 page 11, it
5    is claimed that L1 contains low PV air. But (a) I am completely unable to deduce these two statements from Fig. 3b. Instead
of having a continuous gray shading, can you map only a limited number of gray nuances? Also can you show only positive
PV values? (b) Also isn't it contradictory that L1 and R1 have air masses with different properties if both represent the lobe
region? I guess this is the main problem of the manuscript as it is not precise enough to distinguish R1 and the pouch region,
so that in many places the reader is confused between the two regions.

10    As indicated in the response to the previous comment, L1 and R1 represent separate regions, and so there is no contradiction
on page 10. The PV shading has been changed to make the pv features easier to distinguish.

3. Concerning the non-divergent lobe transport, line 31, page 15, you cannot claim that 'divergence has little impact on
manifold structure' since, precisely, the non-divergent alone cannot explain the size of the lobe! Actually, it is quite interesting
to see that the lobes almost completely disappear (contrary to what you state) and are parallel to contours of the water vapor
15    mixing ratio. This should be discussed as it means that the velocity divergence is responsible of the convergence of the material
lines towards the center of the cyclone and of the inward transport of dry air. On the contrary, non-divergent motions seem
related to a transport barrier so that no air from outside can enter inside the vortex.

The non-divergent flow certainly influences the size of the lobes, and is responsible for the convergence of the material lines.
Though the size of the lobes is greatly reduced, the number of lobes does not change. We have elaborated this discussion on
20    page 15 and emphasized the role of convergence in the inward advection.

4. Last statement page 16 is incorrect: I am unable to see a quick drop in relative humidity outside the shear sheath. Values
are still quite high (r> 85%) at radius r= 2.5.

This statement has been eliminated.

5. Your Okubo-Weiss definition does not take into account the divergent part of the flow. It has to be taken into account
in addition to the shearing effects since it brings material inwards, even if there is a 'shear sheath'. This is quite important to
discuss, as even hyperbolicity is ill-defined in this context.

5    The magnitude of vorticity is typically an order of magnitude greater than divergence, and has very little effect on the OW
values. The following figure shows the OW parameter without divergence, as in Figure 3(c) in the paper, and with divergence.
The differences are very minor, so we have made no changes to the figures, but have added a comment about the role of
divergence where OW is introduced.

90    Minor comments: 1. Top of page 10. In ECMWF, L1 has a negative circulation. On the contrary, when looking at Fig. 4a,
L1 in WRF simulation is positive. Please comment.

The circulation of L1 in the WRF simulation is actually still negative, and should be clearer with the new color scale.

[Figure]

**Figure 1.** The stable (red, magenta) and unstable (blue, cyan) manifolds are overlaid on the $OW$ field $(s^{-2})$ without divergence in (a) and with divergence in (b) at 18 UTC Sept. 6 at 700 hPa.

2. The color palette is also awful for (relative?) vorticity in Fig.4. Can you use something that would be more practical? Same thing for Fig.3. All panels are kind of 'blurred'. Also in Fig.4, the color of the manifolds is reversed compared to Fig.2.

We have changed the color palette in Figures 3 and 4 to make vorticity features easier to distinguish. We have also switched the red and magenta manifold colors in Figures 4 and 5 to make them consistent with the other figures.

3. Page 16, line 17. It is incorrect to say that 'the strain regions are advected inwards'. There is no advection in here!

We now state that the strain regions are converge inwards, but not into the core.

4. You still do not explain in the text how results of figure 7 are obtained (radial average around the center defined by ...).

We now state that the center location used for the radial averages in Figure 7 are taken from the best-track storm location.

5. Figure 7. You should say that the radius is in degree.

We have added in the caption that the radius is in degrees.